# IceFormer: Accelerated Inference with Long-Sequence Transformers on CPUs

**Yuzhen Mao, Martin Ester, Ke Li**
School of Computing Science, Simon Fraser University
Burnaby, BC V5A 1S6, Canada
{yuzhenm,ester,keli}@sfu.ca

## Abstract

One limitation of existing Transformer-based models is that they cannot handle very long sequences as input since their self-attention operations exhibit quadratic time and space complexity. This problem becomes especially acute when Transformers are deployed on hardware platforms equipped only with CPUs. To address this issue, we propose a novel method for accelerating self-attention at inference time that works with pretrained Transformer models out-of-the-box without requiring retraining. We experiment using our method to accelerate various long-sequence Transformers, including a leading LLaMA 2-based LLM, on various benchmarks and demonstrate a speedup of $2.73 \times - 7.63 \times$ while retaining $98.6\% - 99.6\%$ of the accuracy of the original pretrained models. The code is available on our project website at `https://yuzhenmao.github.io/IceFormer/`.

## 1 Introduction

Transformers (Vaswani et al., 2017) have powered incredible advances in NLP, as exemplified by large language models (LLMs) such as GPT-4 and LLaMA 2. Increasingly LLMs are applied to exceptionally long input sequences, which enables many exciting applications such as long-form content creation, extended conversations, and large document search and analysis (OpenAI, 2023; Anthropic, 2023). While LLMs can be feasibly trained with expensive hardware accelerators (e.g. GPUs), they need to be deployed on commodity devices, which may only be equipped with CPUs.

However, it is currently challenging to deploy LLMs on CPUs due to their high computation cost (Dice & Kogan, 2021). A significant computational bottleneck arises from the self-attention mechanism that is integral to Transformers – both time and space complexity are quadratic in the sequence length. This problem is exacerbated in the context of LLMs, which are often used on very long sequences.

To handle long input sequences, there has been substantial research into reducing the quadratic time complexity of self-attention – these methods are collectively known as *efficient Transformers*. However, many do not meet the needs of LLMs and are therefore difficult to apply to LLMs.

An ideal acceleration method for LLMs should satisfy four criteria: (1) **No retraining** – the method should not require the model to be retrained, given the enormous computational expense of training LLMs; (2) **Generality** – the method should be applicable to a variety of LLMs, rather than just those trained with particular constraints built-in; (3) **High accuracy** – the method should not introduce large approximation errors, since LLMs have many attention layers and so errors from earlier layers can compound; (4) **Fast inference** – the method should achieve fast test-time performance.

Satisfying all these criteria simultaneously is difficult, and to our knowledge no existing methods can do so. For example, Transformers with fixed attention patterns, e.g., Longformer (Beltagy et al., 2020), require retraining the model before they can be used. Reformer (Nikita et al., 2020) requires keys to be normalized – this requirement is not met in most pretrained models. Nyströmformer (Xiong et al., 2021) and LARA (Zheng et al., 2022) do not support causal masks, which are commonly found in LLMs. Low-rank methods such as Performer (Choromanski et al., 2020) introduce substantial approximation errors, especially when they are not retrained/finetuned.

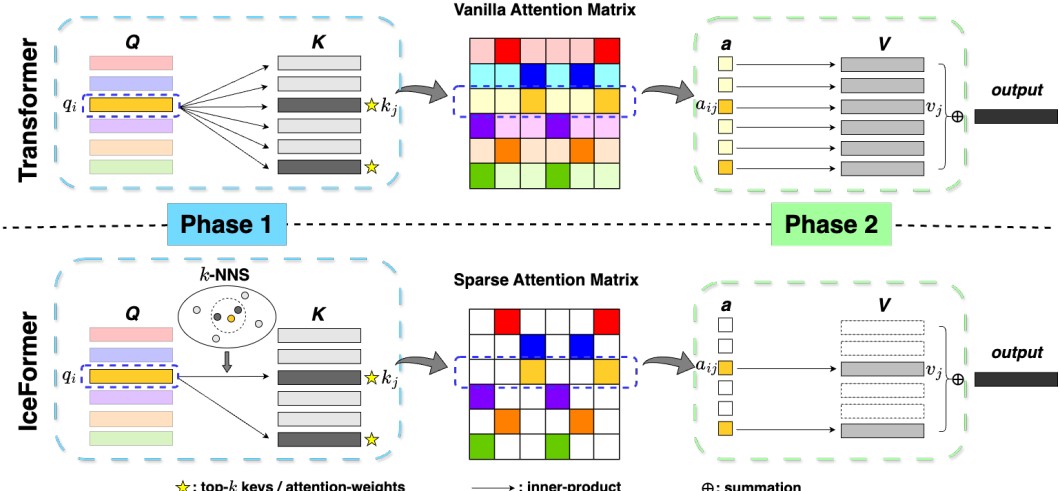

Figure 1: Comparison between Transformer (Vaswani et al., 2017) (top row) and the proposed method, IceFormer (bottom row). We illustrate with one query and $k = 2$ in $k$-NNS. In the two attention matrices presented, the top-2 largest attention weights in each row are represented by a dark color. The remaining attention weights are shown in a pale color in the vanilla attention matrix, and are set to zero (depicted in white) in the sparse attention matrix.

In this paper, we propose an acceleration method, which we dub **IceFormer** due to its ability to be applied directly in frozen models without retraining, that simultaneously satisfies the above four criteria. Specifically, IceFormer (1) does not require retraining, (2) can be applied to most LLMs, (3) can approximate vanilla attention accurately, and (4) achieves significantly faster inference speeds compared to existing methods. We illustrate our method in comparison to the Transformer in Figure 1. As shown, the Transformer computes the attention weights $a_{ij}$ for every possible combination of query $q_i$ and key $k_j$ (Phase 1) and exhaustively enumerates all value vectors $v_j$ for each query (Phase 2). In contrast, our method takes advantage of sparsity of the attention matrix and only computes the highest attention weights and enumerates only the value vectors associated with them.

We conduct experiments on CPUs on the LRA (Tay et al., 2020), ZeroSCROLLS (Shaham et al., 2023), and LongEval (Li et al., 2023) benchmarks. Across all three benchmarks, IceFormer demonstrates substantially faster inference speeds than existing methods while attaining almost no accuracy loss compared to the Transformer. On the LRA benchmark, on average IceFormer achieves a $7.63\times$ speedup relative to the Transformer while retaining $98.6\%$ of its accuracy. Compared to the best efficient Transformer with comparable accuracy for each task, IceFormer is on average $3.04\times$ faster. On the ZeroSCROLLS benchmark, IceFormer achieves a $2.73\times$ speedup on average compared to a leading LLaMA 2-based LLM while retaining $99.6\%$ of its accuracy.

## 2 RELATED WORK

Efficient Transformers can be categorized along two axes: method type and retraining requirement. Along the first axis are sparsity-based methods and low-rank methods. Along the second axis are methods that can and cannot be applied to common pretrained Transformers without retraining.

Sparsity-based methods employ a sparsified attention mechanism to capture global information and integrate it with local attention results. Some approaches aim to improve the space complexity compared to the vanilla attention mechanism without improving the time complexity, e.g., top-$k$ Attention (Gupta et al., 2021). Other approaches aim to improve both, e.g., Sparse Transformer (Child et al., 2019), Longformer (Beltagy et al., 2020), and ETC (Ainslie et al., 2020). A substantial limitation of these models is that the tokens that are attended to are predefined and remain static, which do not adapt to varying input sequences. Because the original attention operation is permitted to attend to any token, these models must be trained with their respective predefined constraints

on tokens to be attended to. Reformer (Nikita et al., 2020) can attend to different sets of tokens for different input sequences by using Locality Sensitive Hashing (LSH) (Andoni et al., 2015) to group tokens into chunks and subsequently attending only to tokens within the same chunk as each query and adjacent chunks. However, Reformer imposes two constraints that are not in the original attention operation: keys must be normalized and queries and keys must be the same. Therefore, Reformer must be trained with these constraints built-in. As a result, these methods cannot be applied to pretrained, non-modified, models directly; instead, the models must be retrained with the required constraints before these methods can be used.

Low-rank methods approximate the attention weight matrix with a low-rank matrix to reduce the quadratic time and space complexity. Examples include Linformer (Wang et al., 2020) and Performer (Choromanski et al., 2020), which decompose the attention weight matrix into a product of tall and wide matrices consisting of learned linear features or random features of the keys and queries, respectively. However, these Transformers typically introduce significant approximation errors because attention weight matrices produced by the original attention operation, especially in the case of long input sequences, typically have high rank. Consequently, models that use these approaches must be trained with low-rank approximations built-in, in order to learn to be robust to the associated approximation errors. As a result, these approaches cannot be applied to pretrained, non-modified, models directly; instead, the models must be retrained with the required approximations before these methods can be used. Other approaches provide more general methodologies that can leverage weights pretrained with standard Transformers without retraining. These Transformers accelerate the execution of the standard attention operation without altering the underlying architecture. Two examples are Nyströmformer (Xiong et al., 2021) and LARA (Zheng et al., 2022), which replace the softmax structure in the self-attention mechanism with the product of separately activated query and key matrices. Nyströmformer utilizes the Nyström method, while LARA combines randomized attention (RA) and random feature attentions (RFA) (Peng et al., 2021) to reconstruct the attention weight matrix. In another example, H-Transformer-1D (Zhu & Soricut, 2021) recursively divides the attention weight matrix into blocks and truncates the small singular values of each off-diagonal blocks. All these approaches leverage low-rank approximations, as opposed to sparsity.

Other works propose hardware-specific optimizations without aiming to improve the computational complexity. Examples include FlashAttention (Dao et al., 2022), which optimizes reads and writes between levels of GPU memory, and H2O (Zhang et al., 2023), which dynamically retains a balance of recent and heavy hitters tokens by a KV cache eviction policy. These strategies are dependent on implementation and are specific to particular hardware platforms (e.g. GPU).

## 3 NOTATION AND PRELIMINARIES

Mathematically, the attention operation takes three matrices as input, $\mathbf{K} \in \mathbb{R}^{m \times d}, \mathbf{Q} \in \mathbb{R}^{n \times d}, \mathbf{V} \in \mathbb{R}^{m \times d'}$, which denote keys, queries and values respectively, and outputs a matrix $\mathbf{O} \in \mathbb{R}^{n \times d'}$. Optionally, it may also take in a mask as input, $\mathbf{S} \in \mathbb{R}^{n \times m}$, whose entries are either 0 or 1. The $i$th rows of $\mathbf{K}, \mathbf{Q}, \mathbf{V}$ and $\mathbf{O}$, denoted as $\mathbf{k}_i, \mathbf{q}_i, \mathbf{v}_i$ and $\mathbf{o}_i$, represent the $i$th key, query, value and output respectively. The entry of $\mathbf{S}$ in the $i$th row and $j$th column, denoted as $s_{i,j}$, represents whether the $i$th query is allowed to attend to the $j$th key — if it is 1, it would be allowed; if it is 0, it would not be. A common masking scheme is the causal mask, where $s_{i,j}$ is 1 if $i \geq j$ and 0 otherwise. Keys and queries have the same dimension $d$, and each key is associated with a value, and so the number of keys and values is the same and denoted as $m$.

First the attention operation computes the attention weight matrix $\mathbf{A} \in \mathbb{R}^{n \times m}$. Its entry in the $i$th row and $j$th column, denoted as $a_{i,j}$, is computed with the following formula:

$$a_{i,j} = \frac{s_{i,j} \exp\left(\frac{\mathbf{q}_i^\top \mathbf{k}_j}{\sqrt{d}}\right)}{\sum_{j'=1}^{m} s_{i,j'} \exp\left(\frac{\mathbf{q}_i^\top \mathbf{k}_{j'}}{\sqrt{d}}\right)} \tag{1}$$

Then the attention operation combines the values with the attention weights in the following way:

$$\mathbf{o}_i = \sum_{j=1}^{m} a_{i,j} \mathbf{v}_j \tag{2}$$

The attention matrix $\mathbf{A}$ is typically sparse (Nikita et al., 2020; Gupta et al., 2021), i.e., in each row of $\mathbf{A}$, only a few attention weights have significant (large) values, while the majority of the remaining values are close to zero. Suppose we can somehow identify the $k$ unmasked keys that receive the highest attention weights for each query $\mathbf{q}_i$ without computing the attention weights for all keys. Then, the original attention matrix $\mathbf{A}$ can be approximated by only computing the inner product for the identified keys, which can save significant amount of time and computational resource.

## 4    IceFormer: Accelerated Self-Attention for General Keys without Retraining

To build a general-purpose retraining-free acceleration method, our approach must not require modifications to the attention mechanism to change attention patterns or the introduction of new model parameters to capture regularities in the attention patterns. This precludes popular strategies such as attention mechanisms with predefined sparse attention patterns, e.g., (Child et al., 2019; Beltagy et al., 2020; Ainslie et al., 2020), and learned dimensionality reduction of keys and queries, e.g., (Wang et al., 2020; Choromanski et al., 2020).

Consequently, it is difficult to design an acceleration method that exploits known regularities in the attention patterns without imposing the retraining requirement. We therefore aim to design an acceleration method that does not make assumptions on the existence of regularity in the attention patterns. In order to improve on the $O(mn)$ complexity of vanilla attention, we need to adaptively identify the most important keys (i.e., those that receive the highest attention weights) without computing all attention weights. This seems like a chicken-and-egg problem: how can we know which attention weights are highest without comparing them to all the other attention weights?

Remarkably, in the special case of normalized keys, as proposed in Nikita et al. (2020), this can be done by leveraging $k$-nearest neighbour search ($k$-NNS) to identify the $k$ most important keys for each query. This relies on the following mathematical fact, whose derivation is in included in Sect. B.1 of the appendix: if $\|\mathbf{k}_j\|_2 = 1$ for all $j$, $\arg\max_j a_{i,j} = \arg\min_j \|\mathbf{q}_i - \mathbf{k}_j\|_2^2$.

However, this fact only holds when all the keys have the same norm – it is not true when different keys differ in their norms. Intuitively, this is because the norms of keys can modulate the attention weights they receive, all else being equal. So if key A has a larger norm than key B, key A can receive a higher attention weight than key B even if key A is farther from the query than key B. As a result, naïvely applying $k$-NNS in the general case would fail to identify the most important keys.

In this paper, we develop an acceleration method that does not require retraining or impose any constraints on keys. It is both accurate and computationally efficient, and can also work with attention masks that are common in Transformers, such as causal masks. Below we will describe the details.

### 4.1    General Retraining-Free Accelerated Attention

Instead of applying $k$-NNS to the original keys directly, we will first embed the keys and queries into a higher dimensional space. Inspired by Neyshabur & Srebro (2015), we choose the following key and query embedding functions, which we denote as $T_K : \mathbb{R}^d \to \mathbb{R}^{d+1}$ and $T_Q : \mathbb{R}^d \to \mathbb{R}^{d+1}$:

$$T_K(\mathbf{k}_j) = \begin{bmatrix} \mathbf{k}_j/c & \sqrt{1 - \|\mathbf{k}_j\|_2^2/c^2} \end{bmatrix}^\top \tag{3}$$

$$T_Q(\mathbf{q}_i) = \begin{bmatrix} \mathbf{q}_i/\|\mathbf{q}_i\|_2 & 0 \end{bmatrix}^\top \tag{4}$$

where $c \geq \max_{j'} \|\mathbf{k}_{j'}\|_2$ is at least the maximum norm across all keys.

It turns out that the $k$ most important keys can be identified by performing $k$-NNS on the key embeddings using the query embedding. We will show this below:

$$\arg\max_j a_{i,j} = \arg\max_j \mathrm{softmax}_j \left( \left\{ \frac{\mathbf{q}_i^\top \mathbf{k}_{j'}}{\sqrt{d}} \right\}_{j'=1}^m \right) \tag{5}$$

$$= \arg \max_j \frac{\mathbf{q}_i^\top \mathbf{k}_j}{\sqrt{d}} \tag{6}$$

$$= \arg \min_j 1 - 2\mathbf{q}_i^\top \mathbf{k}_j / c \|\mathbf{q}_i\|_2 + 1 \tag{7}$$

$$= \arg \min_j \mathbf{q}_i^\top \mathbf{q}_i / \|\mathbf{q}_i\|_2^2 - 2\mathbf{q}_i^\top \mathbf{k}_j / c \|\mathbf{q}_i\|_2 + \mathbf{k}_j^\top \mathbf{k}_j / c^2 + 1 - \|\mathbf{k}_j\|_2^2 / c^2 \tag{8}$$

$$= \arg \min_j \|\mathbf{q}_i / \|\mathbf{q}_i\|_2 - \mathbf{k}_j / c\|_2^2 + 1 - \|\mathbf{k}_j\|_2^2 / c^2 \tag{9}$$

$$= \arg \min_j \|T_Q(\mathbf{q}_i) - T_K(\mathbf{k}_j)\|_2^2 \tag{10}$$

## 4.2 ACCURATE $k$-NNS FOR ACCELERATED ATTENTION

The problem of $k$-NNS is one of the most well studied problems in theoretical computer science. Many algorithms have been developed, and often significant speedups can be obtained by allowing for mistakes with some probability. Such algorithms are known as randomized algorithms.

In the context of LLMs, the number of attention layers is typically high and so errors from earlier layers can compound. Therefore, it is essential for the $k$-NNS algorithm to achieve high accuracy. Choosing an appropriate $k$-NNS algorithm is therefore crucial.

Most $k$-NNS algorithms are bucketing-based, which places keys into discrete buckets and searches over buckets that contain the query. On the other hand, ranking-based algorithms compares the rankings of different keys relative to the query and searches over highly ranked keys. A bucketing-based algorithm effectively uses a fixed threshold on similarity, and so a variable number (including zero) of keys can meet the threshold; on the other hand, a ranking-based algorithm returns a fixed number of keys, which effectively amounts to choosing a variable threshold on similarity based on the distribution of keys, as shown in Figure 2. An example of a bucketing-based algorithm is locality-sensitive hashing (LSH) (Indyk & Motwani, 1998), and an example of a ranking-based algorithm is Prioritized DCI (Li & Malik, 2017). As shown in Figure 2, LSH hashes each key into a bucket associated with the hash value, whereas Prioritized DCI ranks keys along random directions.

For accelerating attention, we posit that ranking-based algorithms are better suited than bucketing-based algorithms, because attention weights depend on how different keys compare to one another, rather than an absolute evaluation of each key against a fixed threshold. Therefore, ranking-based algorithms can yield better recall of truly important keys.

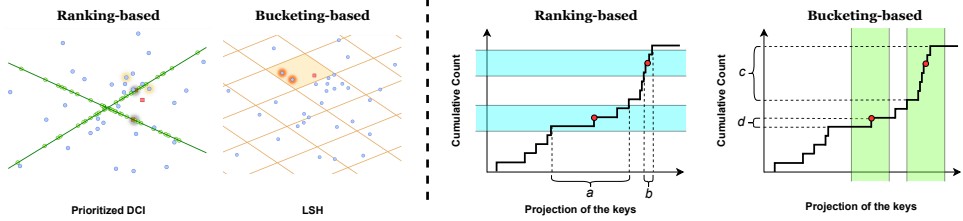

Figure 2: Difference between ranking-based and bucketing-based $k$-NNS. Left: illustration of two $k$-NNS methods, Prioritized DCI (ranking-based) and LSH (bucketing-based). Right: the number of keys whose projections are less than a threshold. Ranking-based algorithms return a fixed number of keys are most similar to the query under projection (shown as a fixed-size row), which effectively filters out points outside a variable-sized window on the projections. Bucketing-based algorithms use a fixed-size window (shown as a fixed-size column) and return all keys whose projections lie within it.

## 4.3 FAST $k$-NNS FOR ACCELERATED ATTENTION

In a Transformer, the keys in an attention layer depend on the output from the preceding attention layer. Therefore, a database needs to be constructed for each attention layer. Therefore, it is important to choose a $k$-NN algorithm that attains both fast construction and querying.

Moreover, in the context of LLMs, many popular models use decoder-only architectures. The attention layers in such architectures use causal masks to prevent the currently generated token to depend on future yet-to-be-generated tokens. Such masked attention is equivalent to excluding the masked out keys from the set of keys the $k$-NNS algorithm operates over. So each time a token is generated, one key becomes unmasked. Instead of constructing a new database each time a token is generated, it is more efficient to add keys incrementally to the database for $k$-NNS.

Fortunately, Prioritized DCI is efficient at both the construction and querying stages. If the number of random projection directions $p$ is nearly as large as the intrinsic dimensionality of the data $d' \geq 1$ and the number of nearest neighbours $k$ to look for is small, Prioritized DCI can return the exact $k$-nearest neighbours for a query with high probability within approximately $\tilde{O}(dk^{p/\tilde{d}}m^{1-p/\tilde{d}})$ time, where $\tilde{O}(\cdot)$ suppresses log factors. Its preprocessing is lightweight, and so only needs $O(dpm)$ time. If we compare this to the computational complexity of vanilla attention of $O(dmn)$, observe that there is no longer a term that depends on $mn$, and so there is no longer the quadratic dependence on sequence length. Later in section 5.1, we also empirically validate the efficiency of Prioritized DCI and found it to be faster than eleven other leading $k$-NNS algorithms.

To support causal masking, we extended the implementation of Prioritized DCI to support incremental database updates. This can be done efficiently, since the data structure consists of sorted lists, so insertions and deletions can be done in $O(\log m)$ time if they are implemented as binary search trees.

# 5 EXPERIMENTS

In this section, we will compare the recall-latency trade-off between different $k$-NNS algorithms and then analyze the performance of IceFormer on the LRA benchmark (Tay et al., 2020), which is a popular benchmark for long-context Transformers (Zhu & Soricut, 2021; Xiong et al., 2021; Zheng et al., 2022). Next we will demonstrate the advantages of IceFormer applied to LLMs with long prompts as input on the ZeroSCROLLS benchmark (Shaham et al., 2023) and the LongEval benchmark (Li et al., 2023). To ensure robustness of results, we used a variety of CPUs for our experiments – we used Intel(R) Core(TM) i7-6850K 6-Core for the LRA experiments, AMD Ryzen 9 5950X 16-Core for the ZeroSCROLLS experiments, and AMD Ryzen 9 5900X 12-Core for the LongEval experiments.

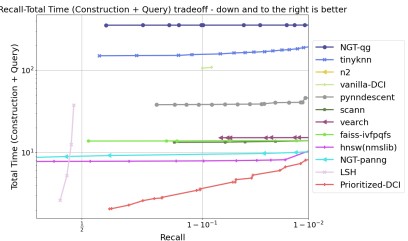

Figure 3: Comparison between twelve $k$-NNS algorithms on fashion-mnist-784 dataset. There are in total 60,000 keys and 10,000 queries with 784 dimensions. The task is to find top-10 closest neighbours from the entire set of keys for every query. X-axis: Average recall across all the queries; Y-axis: Total latency (seconds) including database construction and querying.

## 5.1 DIFFERENT $k$-NNS ALGORITHMS COMPARISON

We compare the recall of true nearest neighbours and total construction and querying time of 12 $k$-NNS algorithms, including Prioritized DCI and the best performing algorithms from ANN benchmarks (Aumüller et al., 2017), on the Fashion MNIST dataset in Figure 3. As shown, Prioritized DCI achieves the best recall-latency trade-off compared to other algorithms, which demonstrates its suitability in our setting, which requires fast construction and querying.

## 5.2 EVALUATION ON LONG RANGE ARENA (LRA) BENCHMARK

**Datasets and Metrics.** LRA consists of five different tasks: ListOps (Nangia & Bowman, 2018), document retrieval (Retrieval) (Radev et al., 2013), text classification (Text) (Maas et al., 2011), CIFAR-10 image classification (Image) (Krizhevsky et al., 2009) and Pathfinder (Linsley et al., 2018). Specifically, all the five tasks consist of sequences with at most 4k tokens. We summarize the dataset information in the appendix C.1 for more details. In this experiment, we follow the train/test splits from Tay et al. (2020) and report the test dataset classification accuracy, average running time of the attention module, and CPU memory usage during inference for each task.

**Baselines.** In addition to the vanilla Transformer, we compare with Nyströmformer (Xiong et al., 2021), H-Transformer-1D (Zhu & Soricut, 2021), LARA (Zheng et al., 2022), Reformer (Nikita et al., 2020), Longformer (Beltagy et al., 2020), Performer (Choromanski et al., 2020), and Linformer (Wang et al., 2020). In order to compare with Reformer, we train a Transformer model with shared **Q** and **K** according to Nikita et al. (2020). For Longformer and Linformer, as they introduce additional parameters, we randomly initialize these parameters when loading the pre-trained weight from the vanilla Transformer. For fair comparisons, we use the LRA evaluation benchmark implemented in PyTorch by (Xiong et al., 2021), and only replace the self-attention module while making other parts of each model exactly the same as the vanilla Transformer.

**Implementation Details.** For each task, we begin by training a base model using GPU with a vanilla Transformer architecture. Then we replace the vanilla attention module with one of the eight efficient attention modules mentioned earlier and directly apply the pre-trained weights for inference. To ensure fair comparison, we adjust the batch size to 1, eliminating the need for a padding mask since our proposed IceFormer automatically ignores padding masks during inference. Note that because of the additional shared-KQ constraint, for the Pathfinder task, our attempts to train a shared-KQ Transformer were unsuccessful. As a result, we have excluded the corresponding results from the subsequent analysis. Additionally, during the inference, we utilize a total of 4 CPU threads. For more comprehensive details, please refer to the appendix C.2.

**Inference Results.** Ideally, the accuracy of the vanilla Transformer (non-shared-KQ) serves as an upper bound for the approximated accuracy of the other seven models (IceFormer (non-shared-KQ), Nyströmformer, H-Transformer-1D, LARA, Longformer, Performer, and Linformer). Similar for the shared-KQ Transformer. Also, the attention module inference time of the vanilla Transformer would be the longest, with other efficient Transformers achieving shorter inference times at the cost of sacrificing prediction accuracy. Table 1 presents the prediction accuracy and inference time of the attention module for each method. The hyper-parameter settings are listed in the appendix C.3. In general, our proposed IceFormer consistently outperforms all efficient Transformers, offering the best accuracy approximation while requiring the least inference time across all five tasks. This demonstrates the generalizability and effectiveness of our model.

Table 1: The performance of vanilla Transformer, and eight approximate attention methods on the LRA benchmarks.

| Method | shared-KQ | ListOps | | Text | | Retrieval | | Image | | Pathfinder | |
|---|---|---|---|---|---|---|---|---|---|---|---|
| | | Acc | Time (s) | Acc | Time (s) | Acc | Time (s) | Acc | Time (s) | Acc | Time (s) |
| Transformer (Vaswani et al., 2017) | ✗ | 0.4255 | 2.9208 | 0.6019 | 0.6933 | 0.6586 | 8.3588 | 0.4132 | 4.9303 | 0.7514 | 0.9620 |
| | ✓ | 0.4145 | 2.9134 | 0.5986 | 0.6603 | 0.6681 | 6.7946 | 0.3844 | 5.9804 | / | / |
| Reformer (Nikita et al., 2020) | ✓ | 0.4121 | 1.4281 | 0.5941 | 0.2288 | 0.6467 | 1.4751 | 0.3726 | 3.6927 | / | / |
| LARA (Zheng et al., 2022) | ✗ | 0.4125 | 0.6146 | 0.5831 | 0.2348 | 0.6401 | 1.8605 | 0.3094 | 2.6720 | 0.7380 | 0.5961 |
| Nyströmformer (Xiong et al., 2021) | ✗ | 0.4128 | 0.7994 | 0.5838 | 0.3542 | 0.6540 | 2.4179 | 0.3754 | 1.7644 | 0.7176 | 0.9927 |
| H-Transformer-1D (Zhu & Soricut, 2021) | ✗ | 0.3265 | 1.9301 | 0.5944 | 0.4811 | 0.5808 | 3.5605 | 0.2286 | 1.2586 | 0.5286 | 0.5708 |
| Longformer (Beltagy et al., 2020) | ✗ | 0.1975 | 0.7406 | 0.5236 | 0.9862 | 0.4918 | 1.0443 | 0.1488 | 0.5451 | 0.5009 | 0.5899 |
| Performer (Choromanski et al., 2020) | ✗ | 0.1975 | 0.6571 | 0.5000 | 0.3327 | 0.4974 | 1.2058 | 0.1345 | 0.6404 | 0.5056 | 0.6395 |
| Linformer (Wang et al., 2020) | ✗ | 0.1975 | 3.1532 | 0.5088 | 1.8912 | 0.4940 | 1.6878 | 0.1064 | 0.7387 | 0.5022 | 1.3141 |
| IceFormer (ours) | ✗ | **0.4153** | **0.3766** | **0.5978** | **0.0921** | **0.6541** | **0.8337** | **0.4046** | **0.5076** | **0.7442** | **0.3058** |
| | ✓ | 0.4124 | 0.4678 | 0.6001 | 0.0903 | 0.6602 | 0.8480 | 0.3752 | 0.9581 | / | / |

**Speed & Accuracy Trade-off.** For IceFormer, increasing the extent of approximation generally improves model efficiency but can lead to a decrease in prediction performance. Here, we study how the extent of approximation affects inference speed and accuracy by varying the number of returned candidates of IceFormer, $k$, from 3 to 10 for each task and present the results in Figure 4. From the figure, we observe that across all tasks, when $k$ becomes larger, IceFormer achieves improved prediction accuracy but becomes less efficient.

**Memory Complexity Analysis.** Table 2 summarizes the maximum memory usage for each method during inference. We employ the same hyper-parameters as in Table 1 and maintain a batch size of 1 to eliminate the need for padding masks. The table reveals that IceFormer consistently exhibits the lowest peak memory usage across all tasks. In comparison to the vanilla Transformer, IceFormer achieves memory savings of up to 0.862 GB.

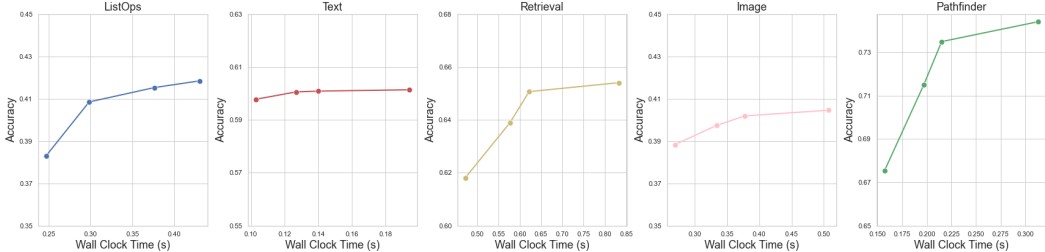

Figure 4: Tradeoff between speed and accuracy as $k$ varies on five LRA tasks. The horizontal axis of each plot is the averaged wall clock time of attention module, and the vertical axis is the model prediction accuracy. Each point corresponds to a value of $k$ in the following set: $\{3, 5, 8, 10\}$.

Table 2: Peak memory usage (GB) on LRA benchmark. The peak memory usage is the total memory usage of the whole program, which includes the memory for the Prioritized DCI database/index.

| Method | shared-KQ | ListOps | Text | Retrieval | Image | Pathfinder |
|---|---|---|---|---|---|---|
| Transformer (Vaswani et al., 2017) | ✗ | 3.729 | 4.327 | 5.031 | 3.778 | 3.926 |
|  | ✓ | 3.631 | 4.265 | 4.877 | 3.740 | / |
| Reformer (Nikita et al., 2020) | ✓ | 3.623 | 3.983 | 4.250 | 3.687 | / |
| LARA (Zheng et al., 2022) | ✗ | 3.584 | 4.129 | 4.566 | 3.772 | 3.943 |
| Nyströmformer (Xiong et al., 2021) | ✗ | 3.478 | 3.982 | 4.375 | 3.463 | 3.845 |
| H-Transformer-1D (Zhu & Soricut, 2021) | ✗ | 3.883 | 4.328 | 4.543 | 3.553 | 3.603 |
| IceFormer (ours) | ✗ | **3.374** | **3.834** | **4.169** | **3.304** | **3.465** |
|  | ✓ | **3.306** | **3.756** | **4.053** | **3.286** | / |

## 5.3 EVALUATION ON LARGE LANGUAGE MODEL (LLM)

We evaluate IceFormer in the LLM setting as well. Specifically, we utilize IceFormer to accelerate the prompt processing process in LLMs. We pick Vicuna-7b-v1.5-16k (Zheng et al., 2023), which is fine-tuned from LLaMA 2 (Touvron et al., 2023) and is one of the top-performing open-source LLMs with a context length up to 16K tokens, for the following experiment. For more comprehensive details including the choice of $k$ in $k$-NNS of IceFormer, please refer to the appendix E.1.

For the following LLM experiments, we do not compare IceFormer with Reformer, LARA and Nyströmformer for the following reasons: Reformer requires keys and queries to be shared, which is not the case in pre-trained LLMs; Longformer only proposed a way to speed up the encoder part of the Transformer, thus cannot be applied to decoder-only LLMs; LARA and Nyströmformer group different tokens into different clusters and so cannot handle causal masks in LLMs, which use decoder-only architectures. All baselines that require retraining (Longformer, Performer and Linformer) are also excluded from the comparison. More details can be found in the appendix E.2.

**ZeroSCROLLS Results.** We compare IceFormer with the vanilla Vicuna-7b-v1.5-16k model and H-Transformer-1D applied to Vicuna-7b-v1.5-16k on the ZeroSCROLLS benchmark (Shaham et al., 2023) which is specifically designed for LLMs and contains ten diverse natural language tasks that require understanding long input contexts, including summarization, question answering, aggregated sentiment classification and information reordering. Each task has a different sequence length varying between 3k and 10k. We measure ZeroSCROLLS scores and latency of the attention module. Table 3 shows that IceFormer achieves up to $3.0\times$ speed-up compared to standard self-attention while attaining at least 99.0% of the vanilla unaccelerated model performance at the same time.

**LongEval Results & Scalability Analysis.** To provide a more comprehensive analysis of Ice-Former's scalability in the LLM setting, we conducted additional experiments on the LongEval benchmark (Li et al., 2023), which is designed to measure long-context performance and consists of two tasks: topic retrieval task with prompt length varying from 3k to 16k, and line retrieval task with prompt length varying from 5k to 16k. In Figure 5, we present the averaged latency of the attention module corresponding to different input prompt length as well as the inference accuracy using the vanilla Vicuna-7b-v1.5-16k model and IceFormer. From the figure, IceFormer can achieve nearly

Table 3: The performance of the vanilla Vicuna-7b-v1.5-16k model, H-Transformer-1D and IceFormer on the ZeroSCROLLS benchmarks. Numbers in parentheses indicate the relative comparison to the vanilla unaccelerated model, denoted as Vicuna-7b-v1.5-16k. We employ the same abbreviations for metric and task names as specified in the original paper (Shaham et al., 2023). We refer interested readers to the original paper for the details.

| Task (#tokens) | Metric | Vicuna-7b-v1.5-16k | H-Transformer-1D | **IceFormer** |
|---|---|---|---|---|
| GvRp (8k) | $R_{geo} \uparrow$ | 11.0 (100%) | 6.8 (61.8%) | **11.0 (100%)** |
| | Time (s) | 5.07 (1.0×) | 4.22 (1.2×) | **1.89 (2.7×)** |
| SSFD (8k) | $R_{geo} \uparrow$ | 13.5 (100%) | 6.3 (46.7%) | **13.5 (100%)** |
| | Time (s) | 5.02 (1.0×) | 4.18 (1.2×) | **1.81 (2.8×)** |
| QMsm (9k) | $R_{geo} \uparrow$ | 16.9 (100%) | 10.7 (63.3%) | **16.8 (99.4%)** |
| | Time (s) | 6.47 (1.0×) | 4.62 (1.4×) | **2.51 (2.6×)** |
| SQAL (8k) | $R_{geo} \uparrow$ | 18.9 (100%) | 7.3 (38.6%) | **18.9 (100%)** |
| | Time (s) | 5.01 (1.0×) | 2.27 (2.2×) | **1.92 (2.6×)** |
| Qspr (5k) | F1 $\uparrow$ | 34.2 (100%) | 6.2 (18.1%) | **34.0 (99.4%)** |
| | Time (s) | 2.03 (1.0×) | 1.70 (1.2×) | **0.89 (2.3×)** |
| Nrtv (10k) | F1 $\uparrow$ | 14.7 (100%) | 2.0 (13.6%) | **14.7 (100%)** |
| | Time (s) | 6.82 (1.0×) | 4.55 (1.5×) | **2.85 (2.4×)** |
| QALT (7k) | AC $\uparrow$ | 48.8 (100%) | 6.8 (13.9%) | **48.6 (99.6%)** |
| | Time (s) | 3.76 (1.0×) | 2.09 (1.8×) | **1.26 (3.0×)** |
| MuSQ (3k) | F1 $\uparrow$ | 18.6 (100%) | 16.9 (90.9%) | **18.5 (99.5%)** |
| | Time (s) | 0.70 (1.0×) | 0.63 (1.1×) | **0.37 (1.9×)** |
| SpDg (7.5k) | ES $\uparrow$ | 42.5 (100%) | 2.9 (6.8%) | **42.3 (99.5%)** |
| | Time (s) | 4.43 (1.0×) | 2.22 (2.0×) | **1.47 (3.0×)** |
| BkSS (7.5k) | $C_{idx} \uparrow$ | 19.5 (100%) | 11.7 (60.0%) | **19.3 (99.0%)** |
| | Time (s) | 4.52 (1.0×) | 2.26 (2.0×) | **1.55 (2.9×)** |
| Avg. (7.5k) | $/ \uparrow$ | 23.9 (100%) | 7.8 (32.5%) | **23.8 (99.6%)** |
| | Time (s) | 4.38 (1.0×) | 2.92 (1.5×) | **1.60 (2.7×)** |

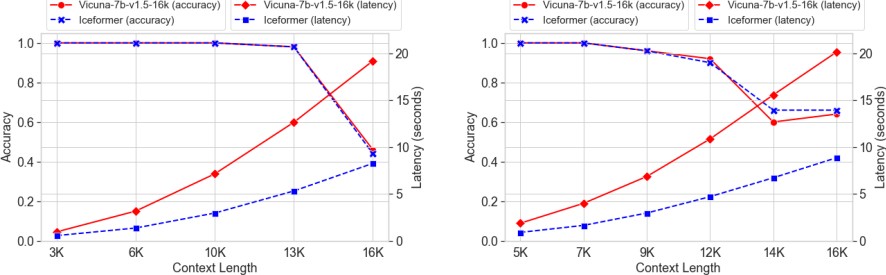

Figure 5: Scalability analysis for IceFormer on the LongEval benchmark. The left figure shows the results of the topic retrieval task; the right figure shows the results of the line retrieval task. X-axis: length of the input prompt; Y-axis (Left): retrieval accuracy; Y-axis (Right): averaged process wall clock time (second) of the attention module.

identical inference accuracy compared with the vanilla Vicuna-7b-v1.5-16k. Notably, as the prompt length increases, there is a corresponding increase in the inference latency for both methods and for both tasks. However, even with very long prompt lengths, IceFormer maintains its scalability and consistently outperforms the vanilla Transformer. Furthermore, as the length of the prompt increases, the difference in the latency between IceFormer and the vanilla Transformer becomes larger, demonstrating the superior scalability and efficiency of IceFormer in the context of LLMs.

## 6 CONCLUSION

In this paper, we present IceFormer, a new method for improving the inference time efficiency of pretrained Transformers on the CPU. Notably, in contrast to other methods, IceFormer does not require retraining, does not require special constraints imposed on the attention mechanism and simultaneously achieves high accuracy and fast inference. These advantages make IceFormer very well-suited to LLM deployment on CPUs, especially when the LLM needs to handle very long sequences as input. The experimental findings on three benchmarks compellingly illustrate the effectiveness of our approach in reducing the quadratic time and space complexity of Transformers both in cases with bi-directional and causal attention mechanisms.

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

# A PSEUDOCODE FOR ICEFORMER

We provide the pseudocode below for IceFormer. For completeness, we also include the pseudocode of Prioritized DCI, which we adapted from (Li & Malik, 2017) and added our modifications. In the following pseudocode below, $T_K$ and $T_Q$ are the key and query embedding functions, respectively. In our implementation, we implement a recursive version of the algorithm, which offers faster performance in practice.

---

**Algorithm 1** IceFormer attention

---

**Require:** A set $S_q$ of $n$ points $q^1, \ldots, q^n \in \mathbb{R}^d$, a set $S_k$ of $n$ keys $k^1, \ldots, k^n \in \mathbb{R}^d$, a set $S_v$ of $n$ values $v^1, \ldots, v^n \in \mathbb{R}^{d'}$, the number of simple indices $m$ that constitute a composite index, the number of composite indices $L$, the number of points to retrieve $k_0$ and the number of points to visit $k_1$ in each composite index

  **function** ICEFORMER_ATTENTION($S_q, S_k, S_v, m, L, k_0, k_1$)

    $\mathbf{O} \leftarrow$ zero matrix $\in \mathbb{R}^{n \times d'}$ with rows $\mathbf{o}_i \in \mathbb{R}^{d'}$

    $S_p \leftarrow$ CONSTRUCT($S_k, m, L$)

    **for** $i = 1$ **to** $n$ **do**

      $S_l \leftarrow$ QUERY($q^i, S_p, k_0, k_1$)

      **for** $j = 1$ **to** $n$ **do**

        **if** $j \in S_l$ **then**

          $\tilde{s}_{ij} \leftarrow \frac{q_i^\top k_j}{\sqrt{d}}$

        **else**

          $\tilde{s}_{ij} \leftarrow 0$

        **end if**

      **end for**

      **for** $j = 1$ **to** $n$ **do**

        **if** $j \in S_l$ **then**

          $\tilde{a}_{i,j} \leftarrow \text{softmax}_j\left(\tilde{s}_{ij}\right) = \frac{\exp\left(\tilde{s}_{i,j}\right)}{\sum_{k \in S_l} \exp\left(\tilde{s}_{i,k}\right)}$

        **else**

          $\tilde{a}_{ij} \leftarrow 0$

        **end if**

      **end for**

      $\mathbf{o}_i \leftarrow \sum_{j \in S_l} \tilde{a}_{i,j} v_j$

    **end for**

    **return** $\mathbf{O}$

  **end function**

---

**Algorithm 2** Data structure construction procedure

---

**Require:** A dataset $D$ of $n$ points $k^1, \ldots, k^n$, the number of simple indices $m$ that constitute a composite index and the number of composite indices $L$

  **function** CONSTRUCT($D, m, L$)

    $\{u_{jl}\}_{j \in [m], l \in [L]} \leftarrow mL$ random unit vectors in $\mathbb{R}^d$

    $\{T_{jl}\}_{j \in [m], l \in [L]} \leftarrow mL$ empty binary search trees or skip lists

    **for** $j = 1$ **to** $m$ **do**

      **for** $l = 1$ **to** $L$ **do**

        **for** $i = 1$ **to** $n$ **do**

          $\overline{k}_{jl}^i \leftarrow \langle T_K(k^i), u_{jl} \rangle$

          Insert $(\overline{k}_{jl}^i, i)$ into $T_{jl}$ with $\overline{k}_{jl}^i$ being the key and $i$ being the value

        **end for**

      **end for**

    **end for**

    **return** $\{(T_{jl}, u_{jl})\}_{j \in [m], l \in [L]}$

  **end function**

---

---

**Algorithm 3** $k$-nearest neighbour querying procedure

---

**Require:** Query point $q$ in $\mathbb{R}^d$, binary search trees/skip lists and their associated projection vectors $\{(T_{jl}, u_{jl})\}_{j \in [m], l \in [L]}$, the number of points to retrieve $k_0$ and the number of points to visit $k_1$ in each composite index

**function** QUERY($q, \{(T_{jl}, u_{jl})\}_{j,l}, k_0, k_1$)
    $C_l \leftarrow$ array of size $n$ with entries initialized to $0 \;\; \forall l \in [L]$
    $\overline{q}_{jl} \leftarrow \langle T_Q(q), u_{jl} \rangle \; \forall j \in [m], l \in [L]$
    $S_l \leftarrow \emptyset \; \forall l \in [L]$
    $P_l \leftarrow$ empty priority queue $\;\; \forall l \in [L]$
    **for** $l = 1$ **to** $L$ **do**
        **for** $j = 1$ **to** $m$ **do**
            $(\overline{p}_{jl}^{(1)}, h_{jl}^{(1)}) \leftarrow$ the node in $T_{jl}$ whose key is the closest to $\overline{q}_{jl}$
            Insert $(\overline{p}_{jl}^{(1)}, h_{jl}^{(1)})$ with priority $-|\overline{p}_{jl}^{(1)} - \overline{q}_{jl}|$ into $P_l$
        **end for**
    **end for**
    **for** $i' = 1$ **to** $k_1 - 1$ **do**
        **for** $l = 1$ **to** $L$ **do**
            **if** $|S_l| < k_0$ **then**
                $(\overline{p}_{jl}^{(i)}, h_{jl}^{(i)}) \leftarrow$ the node with the highest priority in $P_l$
                Remove $(\overline{p}_{jl}^{(i)}, h_{jl}^{(i)})$ from $P_l$ and insert the node in $T_{jl}$ whose key is the next closest to $\overline{q}_{jl}$,
                    which is denoted as $(\overline{p}_{jl}^{(i+1)}, h_{jl}^{(i+1)})$, with priority $-|\overline{p}_{jl}^{(i+1)} - \overline{q}_{jl}|$ into $P_l$
                $C_l[h_{jl}^{(i)}] \leftarrow C_l[h_{jl}^{(i)}] + 1$
                **if** $C_l[h_{jl}^{(i)}] = m$ **then**
                    $S_l \leftarrow S_l \cup \{h_{jl}^{(i)}\}$
                **end if**
            **end if**
        **end for**
    **end for**
    **return** $k$ points in $\bigcup_{l \in [L]} S_l$ that have the maximum inner-product value with $q$
**end function**

---

# B  PROOFS

## B.1  PROOF 1

Here, we provide the full step-by-step derivation of the mathematical equivalence between conducting $k$-nearest neighbour search on normalized keys and identifying the keys that obtain the highest attention weight.

$$\arg\max_j a_{i,j} = \arg\max_j \operatorname{softmax}_j \left( \left\{ \frac{\mathbf{q}_i^\top \mathbf{k}_{j'}}{\sqrt{d}} \right\}_{j'=1}^m \right) \tag{11}$$

$$= \arg\max_j \frac{\mathbf{q}_i^\top \mathbf{k}_j}{\sqrt{d}} \tag{12}$$

$$= \arg\min_j \|\mathbf{q}_i\|_2^2 - 2\mathbf{q}_i^\top \mathbf{k}_j + 1 \tag{13}$$

Since $\|\mathbf{k}_{j'}\|_2 = 1$ for all $j'$, $\|\mathbf{q}_i\|_2^2 - 2\mathbf{q}_i^\top \mathbf{k}_j + 1 = \|\mathbf{q}_i\|_2^2 - 2\mathbf{q}_i^\top \mathbf{k}_j + \|\mathbf{k}_j\|_2^2 = \|\mathbf{q}_i - \mathbf{k}_j\|_2^2$,

$$\arg\max_j a_{i,j} = \arg\min_j \|\mathbf{q}_i\|_2^2 - 2\mathbf{q}_i^\top \mathbf{k}_j + 1 \tag{14}$$

$$= \arg\min_j \|\mathbf{q}_i - \mathbf{k}_j\|_2^2 \tag{15}$$

## B.2  PROOF 2

Here, we provide the full step-by-step derivation of the result in 4.1 establishing the mathematical equivalence between conducting $k$-nearest neighbour search on transformed keys and identifying the

keys that obtain the highest attention weight.

$$\arg\max_j a_{i,j} = \arg\max_j \mathrm{softmax}_j\left(\left\{\frac{\mathbf{q}_i^\top \mathbf{k}_{j'}}{\sqrt{d}}\right\}_{j'=1}^{m}\right) \tag{16}$$

$$= \arg\max_j \frac{\mathbf{q}_i^\top \mathbf{k}_j}{\sqrt{d}} \tag{17}$$

$$= \arg\max_j \mathbf{q}_i^\top \mathbf{k}_j \tag{18}$$

$$= \arg\min_j -2\mathbf{q}_i^\top \mathbf{k}_j \tag{19}$$

$$= \arg\min_j 2 - 2\mathbf{q}_i^\top \mathbf{k}_j/c\|\mathbf{q}_i\|_2 \tag{20}$$

$$= \arg\min_j 1 - 2\mathbf{q}_i^\top \mathbf{k}_j/c\|\mathbf{q}_i\|_2 + 1 \tag{21}$$

$$= \arg\min_j \|\mathbf{q}_i\|_2^2/\|\mathbf{q}_i\|_2^2 - 2\mathbf{q}_i^\top \mathbf{k}_j/c\|\mathbf{q}_i\|_2 + \|\mathbf{k}_j\|_2^2/c^2 + 1 - \|\mathbf{k}_j\|_2^2/c^2 \tag{22}$$

$$= \arg\min_j \mathbf{q}_i^\top \mathbf{q}_i/\|\mathbf{q}_i\|_2^2 - 2\mathbf{q}_i^\top \mathbf{k}_j/c\|\mathbf{q}_i\|_2 + \mathbf{k}_j^\top \mathbf{k}_j/c^2 + 1 - \|\mathbf{k}_j\|_2^2/c^2 \tag{23}$$

$$= \arg\min_j (\mathbf{q}_i/\|\mathbf{q}_i\|_2 - \mathbf{k}_j/c)^\top (\mathbf{q}_i/\|\mathbf{q}_i\|_2 - \mathbf{k}_j/c) + 1 - \|\mathbf{k}_j\|_2^2/c^2 \tag{24}$$

$$= \arg\min_j \|\mathbf{q}_i/\|\mathbf{q}_i\|_2 - \mathbf{k}_j/c\|_2^2 + 1 - \|\mathbf{k}_j\|_2^2/c^2 \tag{25}$$

$$= \arg\min_j \|\mathbf{q}_i/\|\mathbf{q}_i\|_2 - \mathbf{k}_j/c\|_2^2 + \left(0 - \sqrt{1 - \|\mathbf{k}_j\|_2^2/c^2}\right)^2 \tag{26}$$

$$= \arg\min_j \|T_Q(\mathbf{q}_i) - T_K(\mathbf{k}_j)\|_2^2 \tag{27}$$

$$= \arg\min_j \|T_Q(\mathbf{q}_i) - T_K(\mathbf{k}_j)\|_2 \tag{28}$$

## C    More Details on the LRA Experimental Setting

### C.1    Dataset Details

In our LRA experiments, for Retrieval, Text and Pathfinder ($64 \times 64$ version), we directly use the dataset from LRA codebase[1]. Because the original datasets for ListOps and Image only contain short sequences, we generate longer samples for ListOps using the same code from the LRA codebase with 4000 as the maximum length; for Image task, we use a version of the CIFAR-10 dataset super-resolved to $64 \times 64$[2] instead of the original low-resolution $32 \times 32$ CIFAR-10 dataset. We follow the exact same train/test split as the original LRA paper (Tay et al., 2020). The details of the LRA dataset is listed in Table 4.

Table 4: LRA Dataset Details.

| Task | ListOps | Text | Retrieval | Image | Pathfinder |
|---|---|---|---|---|---|
| Max length | 3,991 | 4,000 | 4,000 | 4,096 | 4,096 |
| Avg. length | 2,232 | 1,267 | 3,917 | 4,096 | 4,096 |
| number of classes | 10 | 2 | 2 | 10 | 2 |
| Accuracy by chance | 0.100 | 0.500 | 0.500 | 0.100 | 0.500 |

### C.2    Base Model Configuration

We follow the experimental setup of prior work (Zhu & Soricut, 2021) for training the base model. However, since we were not able to successfully train base Transformer models to satisfactory accuracy on Image and Pathfinder datasets using the original setting, we decreased the number of heads and layers for these two tasks. The details of the base model for each task are outlined in Table 5.

---

[1]https://github.com/google-research/long-range-arena/tree/main
[2]https://www.kaggle.com/datasets/joaopauloschuler/cifar10-64x64-resized-via-cai-super-resolution

Table 5: Configurations of the base models for different tasks.

| Task | ListOps | Text | Retrieval | Image | Pathfinder |
|---|---|---|---|---|---|
| head embedding size | 512 | 512 | 512 | 512 | 512 |
| feed-forward size | 2048 | 2048 | 2048 | 1024 | 1024 |
| number of heads | 8 | 8 | 8 | 4 | 4 |
| number of layers | 6 | 6 | 6 | 4 | 4 |

### C.3 HYPER-PARAMETERS FOR THE BASELINES AND THE PROPOSED METHOD

For LARA and Nyströmformer, we tuned the parameter *num_landmarks* by optimizing over the range {64, 128, 256, 512, 1024}. For H-Transformer-1D, we tuned the parameter *block_size* by optimizing over the range {64, 128, 256, 512, 1024}. For Reformer, we tuned the parameters *num_hash* and *bucket_size*: we considered the values of *num_hash* in range {1, 2, 4} and the values of *bucket_size* in range {64, 128, 256, 512, 1024}. For Longformer, Performer, and Linformer which require retraining, because of their poor performance, we choose hyper-parameter values that result in the least amount of approximation. For IceFormer, we tuned the parameter *top_k* over the range {3, 5, 8, 10, 15, 20}. In general, a larger value for *bucket_size*, *num_landmarks*, *block_size*, or *top_k* indicates less aggressive approximation, meaning that the model performance is closer to that of the vanilla Transformer. We select the values of the hyper-parameters that lead to the best accuracy-time trade-off for each model, and list them in Table 6.

Table 6: Hyper-parameter settings for different methods.

| Method | hyper-parameter | ListOps | Text | Retrieval | Image | Pathfinder |
|---|---|---|---|---|---|---|
| Reformer (Nikita et al., 2020) | num_hash | 1 | 1 | 1 | 1 | / |
| | bucket_size | 512 | 128 | 256 | 1024 | / |
| LARA (Zheng et al., 2022) | num_landmarks | 256 | 256 | 512 | 1024 | 1024 |
| Nyströmformer (Xiong et al., 2021) | num_landmarks | 256 | 256 | 512 | 512 | 1024 |
| H-Transformer-1D (Zhu & Soricut, 2021) | block_size | 1024 | 512 | 1024 | 256 | 1024 |
| Longformer (Beltagy et al., 2020) | attention_window | 2048 | 2048 | 2048 | 2048 | 2048 |
| Performer (Choromanski et al., 2020) | num_rand_features | 2048 | 2048 | 2048 | 2048 | 2048 |
| Linformer (Wang et al., 2020) | num_proj_dim | 2048 | 2048 | 2048 | 2048 | 2048 |
| IceFormer | top_k | 8 | 3 | 10 | 10 | 10 |
| IceFormer (shared-QK) | top_k | 10 | 3 | 10 | 20 | / |

## D ADDITIONAL EXPERIMENTS ON LRA

**Approximation Quality.** In order to assess how well various efficient Transformers approximate the outputs of the vanilla modified attention module, we measure the approximation error by computing the L2-norm of the difference between their attention module outputs and those of the standard vanilla attention module ($\mathbf{o}_i$ in Equation 2). The averaged approximation errors for different efficient Transformers, utilizing the same hyper-parameter settings of Table 6, are summarized in Table 7. As indicated in the table, IceFormer consistently achieves the lowest approximation errors across all LRA tasks, providing further evidence of its approximation efficacy.

Table 7: Quality of the approximation on LRA benchmark. The approximation error of the attention module output is reported for each method across all the tasks.

| Method | shared-KQ | ListOps | Text | Retrieval | Image | Pathfinder |
|---|---|---|---|---|---|---|
| Reformer (Nikita et al., 2020) | ✓ | 3.823 | 3.926 | 5.452 | 2.130 | / |
| LARA (Zheng et al., 2022) | ✗ | 2.395 | 9.456 | 10.025 | 22.066 | 9.261 |
| Nyströmformer (Xiong et al., 2021) | ✗ | 5.758 | 10.269 | 6.523 | 18.789 | 10.442 |
| H-Transformer-1D (Zhu & Soricut, 2021) | ✗ | 6.110 | 10.605 | 5.676 | 53.926 | 12.228 |
| IceFormer (ours) | ✗ | **2.140** | **3.891** | **1.825** | **6.873** | **8.749** |
| | ✓ | 1.562 | 1.686 | 2.499 | 2.127 | / |

**Visualization of Tables 1&2.** The results from Table 1 and Table 2 are visually represented in Figures 6 and 7, respectively.

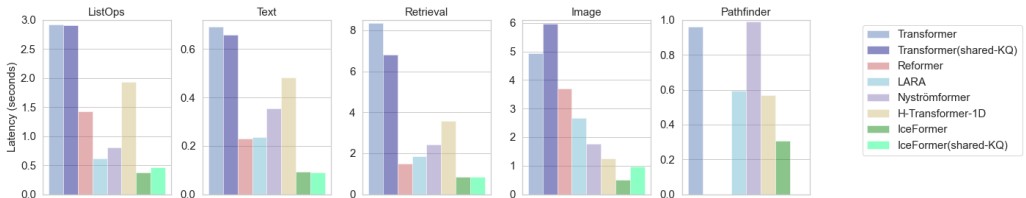

Figure 6: The inference latency of IceFormer and the baselines (vanilla Transformer, Reformer, LARA, Nyströmer, H-Transformer-1D) on the LRA benchmark (the smaller the better).

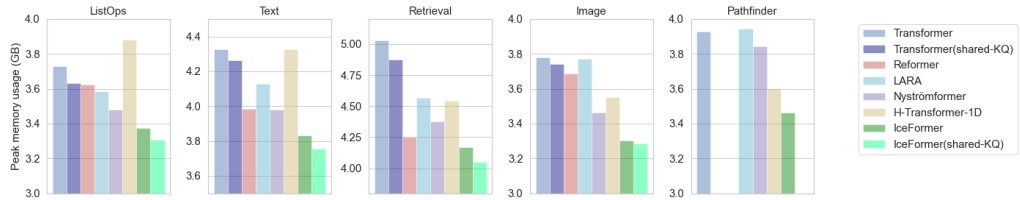

Figure 7: Peak memory usage (GB) on the LRA benchmark. The peak memory usage is the total memory usage of the whole program, which includes the memory for the Prioritized DCI database/index (the smaller the better).

## E   MORE DETAILS ON THE LLM EXPERIMENT

### E.1   LLM EXPERIMENT SETTING

We use vicuna-7b-v1.5-16k as the tested LLM in section 5.3. It contains 32 attention layers, each of which contains 32 attention heads with dimensionality equals to 128. Its maximum input sequence length is 16,384. We observe varying levels of sparsity across different layers of LLMs, and this sparsity remains consistent across different prompts. Therefore, in all the LLMs experiments in section 5.3, we apply IceFormer to approximate relatively sparse layers ranging from the sixteenth to the thirty-first in vicuna-7b-v1.5-16k. This selection encompassed a total of 16 layers, equivalent to half of the total number of layers in the model.

The $k$ in the $k$-NNS of IceFormer for each task of the ZeroSCROLLS benchmark and the LongEval benchmark is defined as:

$$k = \max(\min(\lfloor n * \alpha \rfloor, 50), 30) \tag{29}$$

where $n$ is the number of input tokens, $\lfloor x \rfloor$ is the floor function, and $\alpha$ is a hyper-parameter set by the users. In the ZeroSCROLLS benchmark, we set $\alpha$ equals to 4e-3 for tasks SSFD and QMsm; 5e-3 for tasks GvRp, SQAL, Qspr, Nrtv, MuSQ and BkSS; 6e-3 for tasks QALT and SpDg. In the LongEval benchmark, we set $\alpha$ equals to 5e-3 for all the settings of both two tasks.

### E.2   CAUSAL MASKS AND OTHER INFERENCE-TIME EFFICIENT TRANSFORMERS

In the main paper, we did not compare IceFormer with LARA and Nyströmformer on LLMs. In this section, we elaborate on the problems of causal masks for these two methods.

Most random-feature-based models such as LARA and Nyströmformer group different tokens into different clusters, known as *landmarks*. In order to enable causal masking in these models, not only does the masking need to be applied at the landmark level to prevent the leakage of information from future tokens, an additional set of masks is also required to mask out different numbers of tokens within the same landmark for different queries. The latter is not supported natively and is especially difficult to implement. As a result, it is difficult to apply LARA and Nyströmformer to models that have causal masks.

## F    TEXT OUTPUTS OF ICEFORMER + LLM

In this section, we provide the text outputs of IceFormer when applied to the LLM (vicuna-7b-v1.5-16k) in Figure 8&9. We also include the full information of the input prompts in our supplementary folder.

| Prompt (4k tokens) |
|---|
| A chat between a curious user and an artificial intelligence assistant. The assistant gives helpful, detailed, and polite answers to the user's questions. USER: Please help me to summarize the following:

'There is nothing better than sitting on the couch after a long day. Before that,though, one of the first thoughts running through a person's head probably is, "I wonder what I should eat for supper?" After working for roughly eight hours or longer, no one wants to do is stand over a hot stove for an hour when he or she could be spending that time getting caught up on their favorite Netflix show. Instead of firing up the grill, she pushes a few buttons on a microwave, and voilà! A delicious plate of frozen food, be it pizza bites or a corndog, and cooked to perfection in seconds. Although this is what happens on a regular night in most households, it is a tradition that needs to be put to a stop before it is too late. It's time for people to face the hard and unwelcome facts. Frozen food causes obesity, not just in American, but around the globe.

[...]

Diabetes is caused by the lack of a hormone produced by the pancreas needed to regulate the amount of glucose in the blood. The reason for why Type 1 diabetes occurs in humans is unknown. Type 2 diabetes is frequent in children that are overweight, less active, and often, these children have a parent with diabetes (Hubbard, 2017). "In my career, Type 1 diabetes was a rare disease in children and Type 2 disease didn't exist. And I'm not that old," said Dr. Robin S. Goland said, who has been in practice for almost 25 years and is the co-director of the Naomi Berrie Diabetes Center at Columbia University Medical Center in New York (Saint Louis, 2014). Diabetes could be prevented in children for the most part but this is challenging to do when almost every frozen and pre-packaged food item is seriously unhealthy and goes against nutritional needs.

ASSISTANT: |
| **IceFormer Output Example (k=30)** |
| The article discusses the history and popularity of frozen foods, including the development of quick-freeze methods by Clarence Birdseye and the marketing strategies used by companies like Swanson to promote frozen meals. Frozen foods have become popular due to their convenience, affordability, and availability in grocery stores worldwide. However, the article also notes that frozen meals are often ultra-processed and high in calories, leading to concerns about their impact on health. Additionally, the rise of fast food restaurants has contributed to the increasing consumption of frozen and processed foods in the US. |

Figure 8: Output of IceFormer (Top-30) + LLM with 4k input tokens. We ask the LLM to summarize an article titled "Frozen Food: The World's Favorite Killer".

| Prompt (8k tokens) |
|---|
| A chat between a curious user and an artificial intelligence assistant. The assistant gives helpful, detailed, and polite answers to the user's questions. USER: Please help me to summarize the following:

Rapidly growing population along with digitalization and globalization re-sult in constantly increasing number of internet users. (ITU 2019) Closely relat-ed to this development is the expansion of ecommerce business. Especially in the times of isolation, caused by the Corona crisis, which we currently live in, we spend much more time on the internet than we used to, and this seems to be great oppor-tunity for the growth of online shopping industry. Lockdowns in the countries across the whole world have had considerable influence on the numbers of online purchases. The statistics compiled by the Organisation for Economic Co-operation and Development confirm this assumption showing that the e-commerce retail in the United States and in the United Kingdom rose between the first quarter and the second quarter of 2020 (first lockdown in the UK on the 26th March 2020, in the U.S. on the 19th March 2020 ( from 11.8% to 16.1% respectively from 20.3% to 31.3%. The increasing numbers of online purchases signify higher revenues and at the same time add the importance to this sector.

[...]

If advertisers wish to be successful in their OBA practices, they should also attempt to find out more about users' reasons for being online. According to the interactive advertising model (Rogers and Thorson 2000), advertisements, which address these motives can achieve better results, because consumers tend to pay more attention towards them, and they are also considered to be easier to remem-ber. (Boerman et al. 2017) Consumers are more willing to accept OBA practices, if they feel that they might support them in the pursuit of their objectives. (Kirmani and Campbell 2004)

ASSISTANT: |
| **IceFormer Output Example (k=40)** |
| The rapidly growing population and digitalization have led to an increasing number of internet users, with the ecommerce business expanding in particular during the Corona crisis. Online purchases have risen, and retail volumes have decreased due to lockdowns. Companies are investing more capital in online marketing and advertising, but users often feel irritated by online advertisements, especially when they are not relevant. Online behavioral advertising (OBA) targets consumers based on collected personal information, but it can be effective only if it is highly personalized. However, this can also cause feelings of anxiety and fear, and there have been several scandals related to the misuse of personal data collected online. The importance of the ecommerce sector and the scandals related to online marketing have led to a need to discuss and investigate OBA practices and their influence on consumers. |

Figure 9: Output of IceFormer (Top-40) + LLM with 8k input tokens. We ask the LLM to summarize an article titled "Research of How Online Behavioral Advertising Influences Consumers".

