# OpenReview forum: "IceFormer: Accelerated Inference with Long-Sequence Transformers on CPUs"
_ICLR.cc/2024/Conference — ICLR 2024 poster_

### Official Review · Reviewer_nr9P · 2023-10-27

**Soundness:** 3 good
**Presentation:** 3 good
**Contribution:** 3 good
**Rating:** 6
**Confidence:** 3

**Summary:**

The paper presents a novel approach for adaptively identifying the most important keys for each query using a k-NNS approach with a ranking-based algorithm. This method allows mapping keys and queries to a higher-dimensional space, eliminating the constraint that all keys must have the same norm. Furthermore, the authors extend the implementation of prioritized DCI to support incremental updates to the database.

**Strengths:**

* The paper combines several techniques to accelerate the computation of the attention matrix by pruning the k matrix.
* The proposed approach is comprehensively evaluated on LRA and LLM benchmarks, and it outperforms SOTA approximation methods, demonstrating its effectiveness.

**Weaknesses:**

The adaptation of the k-NNS method may potentially make the computation less parallelizable. It is essential to consider the scalability of the proposed method, even on CPU-based platforms. Leveraging multi-threading and distributed computing for scalability is a relevant concern.

**Questions:**

* It would be valuable if the paper provided insights into the scalability of the proposed method in terms of performance (compute time) on different CPU platforms with varying numbers of cores. How well can the approach be parallelized or scaled up on different hardware configurations?

**Details Of Ethics Concerns:**

No concerns.

---

> ### Author Response · Authors · 2023-11-23
>
> Thank you for your valuable review. Please find our responses as follows.
>
> **Q1: The adaptation of the k-NNS method may potentially make the computation less parallelizable. It is essential to consider the scalability of the proposed method, even on CPU-based platforms. Leveraging multi-threading and distributed computing for scalability is a relevant concern.**
>
> A1: Actually k-NNS does not affect the parallelizability. At the level of tokens, neither vanilla self-attention nor k-NNS parallelizes over them in decoder-only models because they are autoregressive. At the level of heads and token dimensions, both vanilla self-attention and k-NNS can parallelize over them in a straightforward way. In our implementation, we do utilize multi-threading within the k-NNS algorithm. To illustrate the scalability of the proposed method, we added the new experiments as per the reviewer’s suggestions and present the results below.
>
> **Q2: It would be valuable if the paper provided insights into the scalability of the proposed method in terms of performance (compute time) on different CPU platforms with varying numbers of cores. How well can the approach be parallelized or scaled up on different hardware configurations?**
>
> A2: We appreciate the reviewer’s insightful suggestion and tested the proposed IceFormer on a different machine. We reran all the experiments of ZeroSCROLLS benchmark on an AMD Ryzen 9 5950X CPU with 16 CPU cores (where we used 32 threads in total to take advantage of simultaneous multithreading), and compare to the results in the original paper, which were obtained on an AMD Ryzen 9 5900X CPU with 12 cores (where we used24 threads in total):
>
> |               **Task**               | **GvRp (8k)** | **SSFD (8k)** | **QMsm (9k)** | **SQAL (8k)** | **Qspr (5k)** | **Nrtv (10k)** | **QALT (7k)** | **MuSQ (3k)** | **SpDg (7.5k)** | **BkSS (7.5k)** |
> |:------------------------------------:|:-------------:|:-------------:|:-------------:|:-------------:|:-------------:|:--------------:|:-------------:|:-------------:|:---------------:|:---------------:|
> |                                      |               |               |               |               |               |                |               |               |                 |                 |
> | **AMD Ryzen 9 5900X (12 CPU-cores)** |   (old)       |               |               |               |               |                |               |               |                 |                 |
> | **Vicuna-7b-v1.5-16k**               |   5.39        |   5.75        |   7.11        |   5.12        |   2.49        |   7.64         |   4.17        |   0.70        |   4.72          |   4.77          |
> | **Iceformer**                        |   2.24        |   2.14        |   2.67        |   2.15        |   1.06        |   3.39         |   1.85        |   0.49        |   2.09          |   1.96          |
> | **Speed-up**                         |   2.4x        |   2.7x        |   2.7x        |   2.4x        |   2.3x        |   2.3x         |   2.3x        |   1.4x        |   2.3x          |   2.4x          |
> |                                      |               |               |               |               |               |                |               |               |                 |                 |
> | **AMD Ryzen 9 5950X (16 CPU-cores)** |   (new)       |               |               |               |               |                |               |               |                 |                 |
> | **Vicuna-7b-v1.5-16k**               |   5.07        |      5.02         |   6.47        |   5.01        |   2.03        |   6.82         |   3.76        |   0.70        |   4.43          |   4.52          |
> | **Iceformer**                        |   1.89        |   1.81        |   2.51        |   1.92        |   0.89        |   2.85         |   1.26        |   0.37        |   1.47          |   1.55          |
> | **Speed-up**                         |   2.7x        |   2.8x        |   2.6x        |   2.6x        |   2.3x        |   2.4x         |   3.0x        |   1.9x        |   3.0x          |   2.9x          |
>
>
> As shown by the results in the table above, Iceformer can scale effectively to CPUs with more cores. It is notable that the speed-up of Iceformer applied to Vicuna-7b-v1.5-16k compared to the vanilla Vicuna-7b-v1.5-16k model is larger on a CPU with more cores (16) than the CPU used in for the experiments in the original paper, namely a CPU with 12 cores.

---

### Official Review · Reviewer_d3mB · 2023-10-30

**Soundness:** 2 fair
**Presentation:** 3 good
**Contribution:** 3 good
**Rating:** 6
**Confidence:** 5

**Summary:**

This paper presents a method for accelerating inference of long-sequence attention-based Transformers. The proposed method works with pretrained Transformers and does not require retraining. The critical point is to use Prioritized DCI as the k-NNS algorithm.

**Strengths:**

1. The proposed method does not require retraining the model. The retraining may be expensive and unaffordable.
2. It can be applied to a broad range of LLMs since it has no assumptions on the Transformer architecture.
3. The proposed method provides a Pareto optimal solution in terms of time and task-related performance (e.g., accuracy), compared with the baselines.

**Weaknesses:**

### Major issues
1. The proposed method seems orthogonal to the inference platforms. Why do the authors focus on the CPU settings? What about other inference engines, such as GPUs, TPUs, and other accelerators?
2. The authors discuss many related methods in Section 2. However, only few of them are used as the baselines in empirical comparisons? Could the authors add more baselines? Some methods may need retraining. It is better to list more results even if some baseline needs retraining or specific architectures.
3. Could we apply the proposed method into training?

### Minor issues
1. It is not clear how IceFormer will perform on short and medium length of sequences.
2. Could the authors visualize Tables 1 and 2?
3. What are the limitations and potential negative impact of the proposed method?

**Questions:**

1. What may be the future directions and extensions?
2. The authors mentioned that the code will be released upon acceptance. I do not find implementation in the supplementary material. Is it possible to release it for the reviewers' reference?

---

> ### Author Response · Authors · 2023-11-23
> **Part 1/2**
>
> Thank you for your valuable review. Please find our responses as follows.
>
> **Q1: The proposed method seems orthogonal to the inference platforms. Why do the authors focus on the CPU settings? What about other inference engines, such as GPUs, TPUs, and other accelerators?**
>
> A1: Since our main focus is on the deployment of Transformers to end-user devices, we optimized our method for CPUs, since end-user devices are often not equipped with GPUs.
>
> Other hardware platforms such as GPUs and TPUs are fundamentally different from CPUs, with GPUs and TPUs being able to execute a much larger number of parallel threads than CPUs, but also incurring a much higher communication overhead. Therefore, changes need to be made to take advantage of the GPU/TPU features and avoid the performance penalties specific to GPUs/TPUs. Therefore, the optimization approaches that are applicable to CPUs vs. other platforms are very different. On CPUs, the focus is on maximizing cache efficiency and parallelism, both at the instruction level (SIMD) and the thread level (multi-threading). In contrast, on GPUs, the focus is on effectively utilizing the large number of threads without unnecessary communication and synchronization overhead and optimizing accesses to different levels of the memory hierarchy. There isn’t much documentation or literature on TPUs, so effective optimization approaches are somewhat unknown. As a result, performance optimization for different hardware platforms are typically treated separately in the literature, with some papers focusing specifically on CPU optimization, e.g., [1] and others focusing specifically on GPU optimization, e.g., [2].
>
> While it may be possible to extend our method to GPUs, very different optimization approaches will need to be used and changes to the algorithm may be required to minimize communication overhead between different thread blocks. This is beyond the scope of our paper, and we leave it to future work.
>
> **Q2: The authors discuss many related methods in Section 2. However, only few of them are used as the baselines in empirical comparisons? Could the authors add more baselines? Some methods may need retraining. It is better to list more results even if some baseline needs retraining or specific architectures.**
>
> A2: We added Linformer [3], Performers [4] and Longformer [5] as additional baselines. Please see Table 1 and Table 3 in our updated manuscript for details.
>
> **Q3: Could we apply the proposed method into training?**
>
> A3: Yes, it is possible to apply the proposed method to training. Specifically, the same procedure can be used to speed up the forward pass. We did not focus on this use case, however, because we focused on deployment to end-user devices, whereas training is typically done centrally in the cloud.
>
> **Q4: It is not clear how IceFormer will perform on short and medium length of sequences.**
>
> A4: We showed performance on different input lengths in Figure 5. As shown, IceFormer is consistently faster and achieves comparable accuracy compared to the vanilla Vicuna-7b-v1.5-16k model across all input lengths.
>
>
> **Q5: Could the authors visualize Tables 1 and 2?**
>
> A5: Thanks for the suggestion - we added visualizations to the new manuscript. Please refer to the appendix section D.
>
> **Q6: What are the limitations and potential negative impact of the proposed method?**
>
> A6: One limitation is that the value of $k$ needs to be picked manually for each Transformer to best balance efficiency and accuracy. However, this is a one-time cost paid before deploying the model, so it does not affect the performance of the method on the end-user device.
>
> A potential negative impact is that making Transformer inference faster may encourage the development of more inefficient Transformer architectures, because inference speed using such inefficient Transformer architectures may become more tolerable after applying our method.
>
> **Q7: What may be the future directions and extensions?**
>
> A7: One future direction is on developing an extension that can handle non-standard attention masks, beyond simple causal masks explored in this paper. To do so, keys must be inserted into and possibly deleted from the k-NNS database in the appropriate order, so that keys that are masked out for a query cannot be selected by that query.
>
> **Q8: The authors mentioned that the code will be released upon acceptance. I do not find implementation in the supplementary material. Is it possible to release it for the reviewers' reference?**
>
> A8: The process for releasing the code takes some time, and is unfortunately notready by the rebuttal deadline. But we are happy to answer any questions related to the code.

---

> > ### Author Response · Authors · 2023-11-23
> > **Part 2/2**
> >
> > **References:**
> >
> > [1] Zhou, Yi, et al. "Parallel ant colony optimization on multi-core SIMD CPUs." Future Generation Computer Systems 79 (2018): 473-487.
> >
> > [2] Cook, Chase, et al. "GPU based parallel ising computing for combinatorial optimization problems in VLSI physical design." arXiv preprint arXiv:1807.10750 (2018).
> >
> > [3] Wang, Sinong, et al. "Linformer: Self-attention with linear complexity." arXiv preprint arXiv:2006.04768 (2020).
> >
> > [4] Choromanski, Krzysztof, et al. "Rethinking attention with performers." arXiv preprint arXiv:2009.14794 (2020).
> >
> > [5] Beltagy, Iz, Matthew E. Peters, and Arman Cohan. "Longformer: The long-document transformer." arXiv preprint arXiv:2004.05150 (2020).

---

### Official Review · Reviewer_dxGt · 2023-10-31

**Soundness:** 2 fair
**Presentation:** 1 poor
**Contribution:** 1 poor
**Rating:** 3
**Confidence:** 3

**Summary:**

Transformers, especially large language models (LLMs) like GPT-4, have shown great promise in handling long input sequences. However, their deployment on CPUs is challenging due to the quadratic time and space complexity of the self-attention mechanism. Existing methods to address this either require model retraining, are not generalizable, or compromise accuracy. The proposed method, IceFormer, aims to accelerate inference time without retraining, maintaining accuracy, and offering fast inference across various LLMs. It achieves this by optimizing the self-attention mechanism, leveraging the sparsity of the attention matrix, and using k-Nearest Neighbor Search (k-NNS). Experimental results on multiple benchmarks show that IceFormer efficiently outperforms other methods in both speed and accuracy.

**Strengths:**

- This method can accelerate inference time (only CPU) for pretrained transformers without the need for expensive and time-consuming retraining.
- Unlike some other methods, IceFormer ensures that there is minimal approximation error, crucial for LLMs where errors in initial layers can cascade through subsequent ones.
- Beyond just accuracy, the method also guarantees rapid inference times, making it particularly suitable for CPUs.
- By capitalizing on the sparsity of the attention matrix and utilizing k-Nearest Neighbor Search (k-NNS), IceFormer effectively reduces the computational burden of the self-attention mechanism.

**Weaknesses:**

This paper has multiple concerns for acceptance at ICLR 2024:

1) The most glaring issue is its reliance on outdated methods from 2020 and 2021, with some even referencing the 2017 vanilla transformer. This dated focus suggests a lack of recent advancements in the field. One must ponder why the topic of 'efficient transformers' isn't garnering contemporary attention. Historically, efforts to enhance transformer efficiency via attention layer optimization waned with the introduction of Large Language Models (LLM). Moreover, these methods face challenges in adhering to the scaling-law (https://arxiv.org/abs/2207.10551
).

2) Although the paper touts the efficiency of LLM, it primarily showcases small-scale toy examples or, at best, the 7B LLaMa model. Notably, the latter, when applied to a long context, demands an extensive duration, upwards of several seconds, merely for the attention computation. The practical relevance of such scenarios is debatable.
(Honestly, I'm confusing about the exact meaning of 'wall clock time of attention module'.)

3) Presently, in the realm of cloud-based LLMs, MQA/GQA (https://arxiv.org/pdf/2305.13245.pdf) emerges as the leading approach for attention layer optimization. I would be interested in hearing your views on this.

**Questions:**

included in weaknesses.

---

> ### Author Response · Authors · 2023-11-23
> **Part 1/N**
>
> Thank you for your valuable review. Please find our responses as follows.
>
> **Q1: The most glaring issue is its reliance on outdated methods from 2020 and 2021, with some even referencing the 2017 vanilla transformer. This dated focus suggests a lack of recent advancements in the field. One must ponder why the topic of 'efficient transformers' isn't garnering contemporary attention. Historically, efforts to enhance transformer efficiency via attention layer optimization waned with the introduction of Large Language Models (LLM). Moreover, these methods face challenges in adhering to the scaling-law (https://arxiv.org/abs/2207.10551).**
>
> A1:
>
> 1. It is not true that we only compared to outdated methods. In fact, we compared to LARA, which is from  2022. To our knowledge LARA is the latest method in the literature. We only included the vanilla Transformer as a baseline because prior work did - it is not the comparison of primary interest.
>
> 2. Actually efficient Transformers are garnering contemporary attention – there have been multiple survey papers on the topic in 2022 and 2023 [1,2] and leading researchers from OpenAI working on LLMs have written multiple tutorials on the topic in 2023 [3,4]. So efficient Transformers are actually quite topical and are increasingly important due to the development of LLMs.  The reason why research efforts waned with the introduction of LLMs is not due to the lack of importance, but because existing methods require changing the architecture and retraining, which is not feasible for LLMs due to the extremely high computational expense of doing so. Devising a way of improving the efficiency of attention layers without changing the architecture has historically been difficult, since it requires developing a new computational technique to perform the same attention operation used in vanilla Transformers, as opposed to devising a different attention operation. This is precisely the problem we tackle – our method does not need architectural changes or retraining and approximates the same attention operation used in vanilla Transformers without introducing extra parameters, and yet it can still achieve higher efficiency.
>
> 3. We note that our paper does not propose a new efficient Transformer architecture – instead it proposes a general-purpose technique for accelerating Transformers without changing the architecture. The referenced paper (https://arxiv.org/abs/2207.10551) studied the scaling behaviour of different Transformer architectures. It found that only switch Transformer, GLUTransformer and Funnel Transformer achieve comparable results to vanilla Transformer in terms of scaling. This in fact reinforces the need for our method – our method can be applied to speed up the vanilla Transformer without needing to change the architecture, and so the efficiency gain we achieve is *on top of* the scaling benefits offered by the vanilla Transformer, rather than *in place of* them. Our method can also be applied to speed up the switch Transformer, GLUTransformer and Funnel Transformer, since the changes they make relative to the vanilla Transformer do not affect the self-attention layer, which remains unchanged. Therefore, our method is orthogonal to particular Transformer architectures and can be combined with the Transformer architectures with good scaling behaviour to realize the benefits offered by each.
>
> **Q2: Although the paper touts the efficiency of LLM, it primarily showcases small-scale toy examples or, at best, the 7B LLaMa model. Notably, the latter, when applied to a long context, demands an extensive duration, upwards of several seconds, merely for the attention computation. The practical relevance of such scenarios is debatable. (Honestly, I'm confusing about the exact meaning of 'wall clock time of attention module'.)**
>
> A2:
>
> We respectfully disagree with the reviewer on the characterization of the 7B LLaMa 2 model as "small-scale" or a "toy example". The 7B LLaMa 2 model is in fact considered a *large* language model (LLM) by the original authors [5] and is commonly built upon by other LLMs [6,7,8], so it is far from being a "toy example".
>
> In general, it is not meaningful to speak of latency outside of the context of the latency of the original model that our method is used to accelerate. Originally the model took ~5 seconds, and our method reduced it to ~2 seconds. The original model used Intel MKL to perform the attention operation, which is a highly optimized state-of-the-art implementation of BLAS, so the fact that our method is able to achieve a speedup of ~2.5x relative to MKL is quite significant. Furthermore, our work represents just one step towards this direction – future work may build on our work to achieve further speedups.
>
> “wall clock time of attention module” refers to the length of time it takes in the real world to perform attention.

---

> ### Author Response · Authors · 2023-11-23
> **Part 2/2**
>
> **Q3: Presently, in the realm of cloud-based LLMs, MQA/GQA (https://arxiv.org/pdf/2305.13245.pdf) emerges as the leading approach for attention layer optimization. I would be interested in hearing your views on this.**
>
> A3: MQA/GQA is a promising approach for attention layer optimization and works by reducing the number of query and key heads.  MQA uses a single set of key and value heads for all query heads, whereas GQA utilizes a single set of key and value heads for each group of query heads. Our method is general-purpose and can be applied independently of the architecture –
> in particular, it can be applied regardless of the specific number of key, value and query heads. Hence, it can be applied on top of MQA/GQA to achieve a further speedup on top of the benefits offered by MQA/GQA.As of today, MQA/GQA are still relatively new, so their effectiveness in other Transformer models have not been fully demonstrated. As they gain greater adoption, our method can be applied on top of MQA/GQA to further enhance their efficiency.
>
> **References:**
>
> [1]: Yi Tay, Mostafa Dehghani, Dara Bahri, and Donald Metzler. 2022. Efficient Transformers: A Survey. ACM Comput. Surv. 55, 6, Article 109 (June 2023), 28 pages. https://doi.org/10.1145/3530811
>
> [2]: Quentin Fournier, Gaétan Marceau Caron, and Daniel Aloise. 2023. A Practical Survey on Faster and Lighter Transformers. ACM Comput. Surv. 55, 14s, Article 304 (December 2023), 40 pages. https://doi.org/10.1145/3586074
>
> [3:]{https://lilianweng.github.io/posts/2023-01-27-the-transformer-family-v2/}
>
> [4:]{https://lilianweng.github.io/posts/2023-01-10-inference-optimization/}
>
> [5]: Touvron, Hugo, et al. "Llama 2: Open foundation and fine-tuned chat models." arXiv preprint arXiv:2307.09288 (2023).
>
> [6:]{https://www.mosaicml.com/blog/mpt-7b}
>
> [7:]{https://www.together.ai/blog/redpajama-7b}
>
> [8:]{https://huggingface.co/tiiuae/falcon-7b}

---

### Official Review · Reviewer_ad3y · 2023-10-31

**Soundness:** 3 good
**Presentation:** 2 fair
**Contribution:** 2 fair
**Rating:** 6
**Confidence:** 4

**Summary:**

The paper aims to address the computational challenges associated with the quadratic complexity of self-attention in long sequences. It proposes using k nearest-neighbor search as a method for approximating a sparse attention matrix, thereby reducing the computational cost.

Updated post-rebuttal rating.

**Strengths:**

1. The paper is well-motivated, tackling a pertinent issue in the deployment of large language models.
2. The idea of employing ranking-based algorithms over bucketing-based algorithms presents an interesting potential for complexity reduction.

**Weaknesses:**

1. The paper does not adequately support its claim that Prioritized DCI outperforms LSH, lacking both theoretical and empirical evidence.
2. There is insufficient clarity in the algorithm's implementation details, making it difficult to understand the actual complexity and the mechanics of the proposed method.
3. The evaluation methodology for measuring inference time is not comprehensive. It appears the method is optimized for CPUs but lacks evidence of similar efficacy on GPUs.

**Questions:**

1. Are the baseline results for execution time from original papers or are they reproduced results specific to CPU performance?
2. Could you provide the typical and range of values for k and m used in the experiments?
3. What specific settings were used for CPU inference time evaluation? Why limit the experiments to only four CPU threads? Were any BLAS kernels like MKL used in the implementation?

While the paper is well-motivated and presents a potential approach to an important problem, there are significant issues that undermine its contributions. Specifically, the paper lacks rigorous evidence to support its claims and fails to provide a thorough methodology for evaluating its proposed solution. Further revision is needed to substantiate the claims and provide a more comprehensive understanding of the algorithm and its performance across various hardware.

---

> ### Author Response · Authors · 2023-11-23
> **Part 1/ N**
>
> Thank you for your valuable review. Please find our responses as follows.
>
> **Q1: The paper does not adequately support its claim that Prioritized DCI outperforms LSH, lacking both theoretical and empirical evidence.**
>
> A1: From a theoretical perspective, the query time complexity of Prioritized DCI can be found in the original Prioritized DCI paper [1], which is $O\left(d \max \left(\log n, n^{1-m / d^{\prime}}\right)\right.$. The query time complexity of LSH can be found in the original LSH paper for Euclidean spaces [2], which is $\approx O\left(d n^{1 /(1+\epsilon)^2}\right)$. The epsilon denotes the maximum amount of error that is tolerable, and since it is important to tell apart similar keys that are ranked differently in our use case, epsilon needs to be very small, which makes LSH very slow.
>
> From an empirical perspective, we incorporated LSH into the kNNS comparison experiment detailed in section 5.1, and consequently updated Figure 3 in the new manuscript. As shown, Prioritized DCI substantially outperforms LSH in terms of both efficiency and recall. The original Prioritized DCI paper also included an empirical comparison with LSH in its experiments section. For more details, we kindly refer the reviewer to the literature mentioned above.
>
> **Q2: There is insufficient clarity in the algorithm's implementation details, making it difficult to understand the actual complexity and the mechanics of the proposed method.**
>
> A2: We thank the reviewer for pointing this out. To improve the clarity, we have included the pseudocode for our method in the updated appendix section A in the new manuscript.
>
> **Q3: The evaluation methodology for measuring inference time is not comprehensive. It appears the method is optimized for CPUs but lacks evidence of similar efficacy on GPUs.**
>
> A3: CPU code cannot be run on the GPU and vice versa, so one cannot simply evaluate performance of a CPU implementation on the GPU without creating a separate implementation optimized specifically for GPUs. CPUs and GPUs are fundamentally different hardware platforms, with GPUs being able to execute a much larger number of parallel threads than CPUs, but also incurring a much higher communication overhead. Therefore, changes need to be made to take advantage of the GPU features and avoid the performance penalties specific to GPUs. Therefore, the optimization approaches that are applicable to one platform vs. the other are very different. On CPUs, the focus is on maximizing cache efficiency and parallelism, both at the instruction level (SIMD) and the thread level (multi-threading). In contrast, on GPUs, the focus is on effectively utilizing the large number of threads without unnecessary communication and synchronization overhead and  optimizing accesses to different levels of the memory hierarchy. As a result, performance optimization for CPUs vs. GPUs are typically treated separately in the literature, with some papers focusing specifically on CPU optimization, e.g., [3] and others focusing specifically on GPU optimization, e.g., [4].
>
> Since our main focus is on the deployment of Transformers to end-user devices, we optimized our method for CPUs, since end-user devices are often not equipped with GPUs. While it may be possible to extend our method to GPUs, very different optimization approaches will need to be used and changes to the algorithm may be required to minimize communication overhead between thread blocks. This is beyond the scope of our paper, and we leave it to future work.
>
> **Q4: Are the baseline results for execution time from original papers or are they reproduced results specific to CPU performance?**
>
> A4: As reflected in our paper’s title, i.e., “Accelerated Inference with Long-Sequence Transformers on CPUs”, we focus on deployment of Transformers to end-user devices equipped only with CPUs, so the baseline are the results of running the official code accompanying the original papers or widely-used HuggingFace implementations on the CPU.
>
> **Q5: Could you provide the typical and range of values for k and m used in the experiments?**
>
> A5: We list the range of values for $k$ in Table 6 in the appendix for the LRA benchmark, and in Eq. 29 under Section E of the appendix, for the LLM experiments. The sequence length $m$ of the LRA benchmark can be found in Table 4 in the appendix. The $m$ of the LLM experiments can be found in Table 3 and Figure 5.

---

> > ### Author Response · Authors · 2023-11-23
> > **Part 2/N**
> >
> > **Q6: What specific settings were used for CPU inference time evaluation? Why limit the experiments to only four CPU threads? Were any BLAS kernels like MKL used in the implementation?**
> >
> > A6: For the specific settings that were used for CPU inference time evaluation, please refer to the first paragraph of Sec. 5.
> >
> > There was no particular reason to limit the experiments to any particular number of CPU threads – we simply used the hardware that was available at the time we did the experiments and set the number of threads to a fixed value from then on to maintain consistency in our comparisons. We actually used 24 CPU threads in the LLM experiments (because the CPU, AMD Ryzen 9 5900X, has 12 cores). In response to the reviewer’s question, we also reran our experiments with 32 CPU threads on an AMD Ryzen 9 5950X 16-core CPU, and found the speedup Iceformer achieves relative to the vanilla Vicuna-7b-v1.5-16k model to be even more significant compared to the results with 24 CPU threads. Detailed results are available in the table below.
> >
> > Yes, in all our experiments, we used BLAS kernels (from Intel MKL). It's important to note that all the baseline methods we compared to also used the same BLAS kernels from Intel MKL since torch.matmul() used in all the baselines leverages BLAS for acceleration.
> >
> > |               **Task**               | **GvRp (8k)** | **SSFD (8k)** | **QMsm (9k)** | **SQAL (8k)** | **Qspr (5k)** | **Nrtv (10k)** | **QALT (7k)** | **MuSQ (3k)** | **SpDg (7.5k)** | **BkSS (7.5k)** |
> > |:------------------------------------:|:-------------:|:-------------:|:-------------:|:-------------:|:-------------:|:--------------:|:-------------:|:-------------:|:---------------:|:---------------:|
> > |                                      |               |               |               |               |               |                |               |               |                 |                 |
> > | **AMD Ryzen 9 5900X (12 CPU-cores)** |   (old)       |               |               |               |               |                |               |               |                 |                 |
> > | **Vicuna-7b-v1.5-16k**               |   5.39        |   5.75        |   7.11        |   5.12        |   2.49        |   7.64         |   4.17        |   0.70        |   4.72          |   4.77          |
> > | **Iceformer**                        |   2.24        |   2.14        |   2.67        |   2.15        |   1.06        |   3.39         |   1.85        |   0.49        |   2.09          |   1.96          |
> > | **Speed-up**                         |   2.4x        |   2.7x        |   2.7x        |   2.4x        |   2.3x        |   2.3x         |   2.3x        |   1.4x        |   2.3x          |   2.4x          |
> > |                                      |               |               |               |               |               |                |               |               |                 |                 |
> > | **AMD Ryzen 9 5950X (16 CPU-cores)** |   (new)       |               |               |               |               |                |               |               |                 |                 |
> > | **Vicuna-7b-v1.5-16k**               |   5.07        |       5.02        |   6.47        |   5.01        |   2.03        |   6.82         |   3.76        |   0.70        |   4.43          |   4.52          |
> > | **Iceformer**                        |   1.89        |   1.81        |   2.51        |   1.92        |   0.89        |   2.85         |   1.26        |   0.37        |   1.47          |   1.55          |
> > | **Speed-up**                         |   2.7x        |   2.8x        |   2.6x        |   2.6x        |   2.3x        |   2.4x         |   3.0x        |   1.9x        |   3.0x          |   2.9x          |

---

> > > ### Author Response · Authors · 2023-11-23
> > > **Part 3/3**
> > >
> > > **Q7: While the paper is well-motivated and presents a potential approach to an important problem, there are significant issues that undermine its contributions. Specifically, the paper lacks rigorous evidence to support its claims and fails to provide a thorough methodology for evaluating its proposed solution. Further revision is needed to substantiate the claims and provide a more comprehensive understanding of the algorithm and its performance across various hardware.**
> > >
> > > A7: We appreciate the reviewer's insightful suggestions. As we emphasized in the introduction, our focus is on the *deployment* of Transformers to end-user devices not equipped with hardware accelerators. We evaluated our method on various hardware platforms, such as Intel(R) Core(TM) i7-6850K, AMD Ryzen 9 5900X and AMD Ryzen 9 5950X. We would like to clarify that the information requested, including theoretical/empirical evidence, pseudocode, and hyperparameters of our experiment, is available in the referenced literature and our original paper. Regarding the evaluation of performance on various hardware platforms, we have to point out that the specific implementations on different hardware platforms are very different and involve different optimization considerations, and our focus, as reflected in the paper's title, is on accelerated inference using Long-Sequence Transformers specifically on CPUs. Nevertheless, there is certainly potential for extending our method to other types of hardware platforms, but it is beyond the scope of our paper and we leave it to future work.
> > >
> > > **References:**
> > >
> > > [1] Li, Ke, and Jitendra Malik. "Fast k-nearest neighbour search via prioritized DCI." International conference on machine learning. PMLR, 2017.
> > >
> > > [2] Andoni, Alexandr, and Piotr Indyk. "Near-Optimal Hashing Algorithms for Approximate Nearest Neighbor in High Dimensions." 2006 47th Annual IEEE Symposium on Foundations of Computer Science (FOCS'06).
> > >
> > > [3] Zhou, Yi, et al. "Parallel ant colony optimization on multi-core SIMD CPUs." Future Generation Computer Systems 79 (2018): 473-487.
> > >
> > > [4] Cook, Chase, et al. "GPU based parallel ising computing for combinatorial optimization problems in VLSI physical design." arXiv preprint arXiv:1807.10750 (2018).

---

> ### Comment · Reviewer_ad3y · 2023-12-04
>
> Thank the authors for the rebuttal. With the clarified scope of this paper being CPU-oriented, I am okay with the revision and will raise my score.

---

### Official Review · Reviewer_5Bmg · 2023-11-01

**Soundness:** 3 good
**Presentation:** 3 good
**Contribution:** 3 good
**Rating:** 6
**Confidence:** 3

**Summary:**

In this paper, the authors present IceFormer, a novel method for improving the inference efficiency of pretrained Transformers on the CPU without requiring re-training. Leveraging the insight that k most important keys can be identified by performing k-NNS on the key
embeddings using the query embedding, IceFormer employs the Prioritized DCI algorithm to identify the top-k keys. The evaluations on three benchmarks illustrate the effectiveness of this approach in reducing the quadratic time and space complexity of vanilla Transformers with competitive performance compared to other efficient-attention variants.

**Strengths:**

1. The paper sheds new insights on an important problem -- choosing the top-k keys in sparse attention for inference acceleration. The authors show that the exact kNNS algorithm is critical for the success of sparse attention.
2. The paper is well-written and easy to follow.
3. The evaluation section shows strong performance compared to other efficient attention works.

**Weaknesses:**

1. The use of Prioritized DCI k-NNS algorithm needs more experimental or theoretical justification. The authors claim "ranking-based algorithms is better aligned with how attention weights", if so, how would other ranking-based algorithms perform? On top of that, the authors show an evaluation of different kNNS algorithms on fashion-mnist-784 dataset in Section 5.1. It would be better to show the exact setup (eg. model architecture) they used, and compare them on a few more tasks (for instance, text generation, token classification).
2. Fig.5's y-axis needs to be annotated with units (especially for latency).

**Questions:**

1. Why is CPU the target platform for IceFormer? Are there CPU-specific architectural optimizations, on top of the time/space saved due to sparse attention?
2. I hope to clarify with the authors: in the long-context evaluation (Fig.5), is the baseline (vanilla Transformer) referring to the Vicuna-7b-v1.5-16k model, or the original Transformer model?
3. This paper describes a novel method for improving the inference efficiency and evaluates on CPU. But it seems that this method could potentially also apply to GPU, which is more often used. Are there specific reasons/constraints to use CPU?

---

> ### Author Response · Authors · 2023-11-23
>
> Thank you for your valuable review. Please find our responses as follows.
>
> **Q1: The use of Prioritized DCI k-NNS algorithm needs more experimental or theoretical justification. The authors claim "ranking-based algorithms is better aligned with how attention weights", if so, how would other ranking-based algorithms perform?**
>
> A1: There have only been two ranking-based algorithms, vanilla DCI [1] and Prioritized DCI [2]. To respond to the reviewer’s question, we have incorporated both algorithms into the k-NNS algorithm comparison  detailed in Section 5.1, and updated Figure 3 accordingly. As shown in the figure, Prioritized DCI is significantly more efficient than vanilla DCI.
>
> **Q2: On top of that, the authors show an evaluation of different kNNS algorithms on fashion-mnist-784 dataset in Section 5.1. It would be better to show the exact setup (eg. model architecture) they used, and compare them on a few more tasks (for instance, text generation, token classification).**
>
> A2: For the evaluation of different k-NNS algorithms, we followed the evaluation protocol of the ANN benchmarks (https://github.com/erikbern/ann-benchmarks). More details can be found on the website of ANN benchmarks. ANN benchmarks is a general benchmark for k-NNS algorithms, and is not specific to a particular model architecture. It also includes the references for each k-NNS algorithm tested.
>
> As per the reviewer’s request, we compare the second best k-NNS algorithm as measured by ANN benchmarks in Figure 3, on two tasks, the text classification task in the LRA benchmark and the GvRp (text generation) task in the ZeroSCROLLS. The second best k-NNS algorithm is the hnsw(nmslib) algorithm. The latency for hnsw(nmslib) is 1.06 seconds on the text classification task and 15.76 seconds on the GvRp task, significantly slower than the vanilla self-attention (0.69 seconds and 5.39 seconds) and Iceformer (0.09 seconds, and 2.24 seconds). This is because evaluating each attention layer requires constructing a new k-NNS database, which takes a long time forhnsw(nmslib) and many other k-NNS algorithms. On the other hand, constructing a new k-NNS database is much faster with Prioritized DCI. This result is consistent with our k-NNS comparison results in Figure 3 and the literature [3, 4].
>
> **Q3: Fig.5's y-axis needs to be annotated with units (especially for latency).**
>
> A3: We followed the reviewer’s suggestion and added the units for latencyin Figure 5 of the updated manuscript.
>
> **Q4: Why is CPU the target platform for IceFormer? Are there CPU-specific architectural optimizations, on top of the time/space saved due to sparse attention?**
>
> A4: Since our main focus is on the deployment of Transformers to end-user devices, we optimized our method for CPUs, since end-user devices are often not equipped with GPUs. We used various CPU-specific optimizations to minimize the wall-clock time, namely instruction-level parallelism (SIMD), bitwise operations and cache-optimized data structures.
>
> **Q5: I hope to clarify with the authors: in the long-context evaluation (Fig.5), is the baseline (vanilla Transformer) referring to the Vicuna-7b-v1.5-16k model, or the original Transformer model?**
>
> A5: We thank the reviewer for pointing this out. The baseline (vanilla Transformer) refers to the vanilla Vicuna-7b-v1.5-16k model. We have fixed the labels in Table 3 and Figure 5.
>
> **Q6: This paper describes a novel method for improving the inference efficiency and evaluates on CPU. But it seems that this method could potentially also apply to GPU, which is more often used. Are there specific reasons/constraints to use CPU?**
>
> A6: While there is certainly potential to extend our method to GPUs, very different optimization approaches will need to be used and changes to the algorithm may be required to minimize communication overhead between different thread blocks. This is beyond the scope of our paper, and we leave it to future work. As mentioned earlier, our implementation uses CPU-specific optimizations, such as instruction-level parallelism (SIMD), bitwise operations and cache-optimized data structures. Different optimizations will need to be used to minimize the wall-clock time on GPUs.
>
> **References:**
>
> [1]: Li, Ke, and Jitendra Malik. "Fast k-nearest neighbour search via dynamic continuous indexing." International conference on machine learning. PMLR, 2016.
>
> [2]: Li, Ke, and Jitendra Malik. "Fast k-nearest neighbour search via prioritized DCI." International conference on machine learning. PMLR, 2017.
>
> [3]: Echihabi, Karima, et al. "The lernaean hydra of data series similarity search: An experimental evaluation of the state of the art." arXiv preprint arXiv:2006.11454 (2020).
>
> [4]: Echihabi, Karima, et al. "Return of the lernaean hydra: Experimental evaluation of data series approximate similarity search." arXiv preprint arXiv:2006.11459 (2020).

---

> > ### Comment · Reviewer_5Bmg · 2023-12-05
> >
> > Thank you for the detailed response and clarification to my questions. I have no further questions on this submission.

---

### Comment · Area_Chair_2hDq · 2023-11-22

Dear all,

The author-reviewer discussion period is about to end.

@authors: If not done already, please respond to the comments or questions reviewers may further have. Remain short and to the point.

@reviewers: Please read the author's responses and ask any further questions you may have. To facilitate the decision by the end of the process, please also acknowledge that you have read the responses and indicate whether you want to update your evaluation.

You can update your evaluation positively (if you are satisfied with the responses) or negatively (if you are not satisfied with the responses or share other reviewers' concerns). Please note that major changes are a reason for rejection.

You can also keep your evaluation unchanged. In this case, please indicate that you have read the responses, that you do not have any further comments and that you keep your evaluation unchanged.

Best regards,
The AC

---

### Author Response · Authors · 2023-11-23
**General Response**

We thank the reviewers for their reviews. Below is a one-sentence summary of our answer to each of the key questions raised by the reviewers. Details are in the individual response to each reviewer.

**Q1: (R 5Bmg, ad3y, d3mB) Why is the CPU the target platform for IceFormer?**

A1: Since our main focus is on the deployment of Transformers to end-user devices, we optimized our method for CPUs, since end-user devices are often not equipped with GPUs.

**Q2: (R 5Bmg) How would other ranking-based algorithms perform compared to Prioritized DCI?**

A2:  We have added additional experiments that show the performance of Prioritized DCI significantly outperforms the only other ranking-based algorithm.

**Q3: (R 5Bmg) The long-context evaluation (Fig.5), is the baseline (vanilla Transformer) referring to the Vicuna-7b-v1.5-16k model, or the original Transformer model?**

A3:  It refers to the Vicuna-7b-v1.5-16k model. We have clarified this in the manuscript.

**Q4: (R ad3y) There is insufficient clarity in the algorithm's implementation details, making it difficult to understand the actual complexity and the mechanics of the proposed method.**

A4: We have included the pseudocode of our method in the updated appendix section A of the manuscript.

**Q5: (R d3mB) Could the authors add more baselines? Some methods may need retraining. It is better to list more results even if some baseline needs retraining or specific architectures.**

A5: We have added three baseline methods that require retraining, namely Linformer, Performer and Longformer.

**Q6: (R nr9P) How well can the approach be parallelized or scaled up on different hardware configurations?**

A6:  We have added additional experiments that show the scalability of our method to more CPU cores.

**Q7: (R d3mB) Could we apply the proposed method to training?**

A7: Yes, it is possible to apply the proposed method to training, which can speed up the forward pass.

**Q8: (R d3mB) What may be the future directions and extensions?**

A8: One future direction is to develop extensions that handle non-standard attention masks, beyond causal masks.

---

### Meta-Review · Area_Chair_2hDq · 2023-12-10

**Metareview:**

Overall, the reviewers tend towards acceptance (6-6-3-6-6), even if none of them is strongly in favour either. The paper proposes a novel approach to accelerate self-attention at inference time that works with pretrained models without requiring retraining. The reviewers note the importance of the problem and the original of the contribution. The authors-reviewers discussion has been constructive and has led to a number of improvements to the paper, including a better presentation and a more thorough empirical evaluation. Only Reviewer dxGt (3) remains unconvinced by the contribution. Unfortunately, no discussion has taken place after the authors' reply. Given this situation and the otherwise positive reviews, I recommend acceptance. I also encourage the authors to address the remaining concerns the reviewers may have in the final version of the paper.

**Justification For Why Not Higher Score:**

Only slighlty above the acceptance the threshold.

**Justification For Why Not Lower Score:**

The counter-reply of the authors to Reviewer dxGt is good.

---

### Decision · Program_Chairs · 2024-01-16

Accept (poster)